# Understanding the Generalization of Blind Image Quality Assessment: A Theoretical Perspective on Multi-level Quality Features

## Abstract

Due to the high annotation costs and relatively small scale of existing Image Quality Assessment (IQA) datasets, attaining consistent generalization remains a significant challenge for prevalent deep learning (DL)-based IQA methods. Although it is widely believed that quality perception information primarily resides in low-level image features, and that effective representation learning for the multi-level image features and distortion information is deemed crucial for the generalization of the Blind IQA (BIQA) methods, the theoretical underpinnings for this belief still remain elusive. Therefore, in this work, we investigate the role of multi-level image features in the generalization and quality perception ability of the CNN-based BIQA models from a theoretical perspective. For the role of low-level features, in Theorem 1, we innovatively derive an upper bound of Rademacher Average and the corresponding generalization bound for the CNN-based BIQA framework under distribution invariance in training and test sets, which indicates that the generalization ability tends to be reduced as the level of quality features increases, demonstrating the value of low-level features. In addition, under distribution shifts, a much tighter generalization bound is proposed in Theorem 2, which elucidates the theoretical impact of distributional differences between training and test sets on generalization performance. For the role of high-level features, in Theorem 3, we prove that BIQA networks tend to possess higher Betti number complexity by learning higher-level quality features. This indicates a larger representation power with smaller empirical errors, highlighting the value of high-level features. The three proposed Theorems can provide theoretical support for the enhanced generalization in existing BIQA methods. Furthermore, these theoretical findings reveal an inherent tension between robust generalization and strong representation power in BIQA networks, which inspires us to explore effective strategies to reduce empirical error without compromising the generalization ability. Extensive experiments validate the reliability and practical value of our theoretical findings.

## 1 Introduction

Image quality assessment (IQA) plays a crucial role in optimizing visual experiences across different domains, including image denoising (Tian et al., 2020), restoration (Cui et al., 2023), and generation (Elasri et al., 2022). Its primary objective is to develop algorithms that can accurately predict image quality scores consistent with subjective human ratings (i.e., Mean Opinion Scores, MOS). Based on the availability of reference information, existing IQA methods can be classified into Full-Reference IQA (FR-IQA), Reduced-Reference IQA (RR-IQA), and Blind IQA (BIQA) (Zhai & Min, 2020). Among these, BIQA exhibits broader applicability due to its independence from the reference image (Simeng et al., 2023; Feng et al., 2021). Traditional BIQA methods aim to quantify the perceptual quality by manually extracting or selecting valid statistical features from the distorted images (Jiang et al., 2017; Liu et al., 2020; Zhou et al., 2017). These methods have shown promising results in evaluating images with synthesized distortions, while they perform poorly in authentically distorted scenarios. Therefore, numerous studies have springed up on Deep Learning (DL)-based BIQA methods for authentically distorted images, owing to their success in fusing discriminative features in visual domains (Ke et al., 2021; Madhusudana et al., 2022; Simeng et al., 2023).However, the training process of these methods requires a large amount of data to prevent overfitting, while there

exists no large database of authentically distorted images nowadays due to the expensive annotation process, leading to poor generalization (Prabhakaran & Swamy, 2023; Yue et al., 2022).

To address this issue, one of the most intuitive and effective ideas is to explore more effective network architectures and training paradigms for the IQA task (Liu et al., 2017; Lin et al., 2020; Ma et al., 2017; Zhou et al., 2022). These studies have revealed that the generalization can be improved by learning effective visual representations based on complex structural designs or additional knowledge injection, but this comes at the cost of higher training costs. Therefore, the more economical design schemes are valuable and worthy of investigation, which aim to attain a significant boost in generalization performance by only trading off a marginal efficiency loss without extra training modules or data annotations. Given this, and in light of the fact that quality perception information primarily resides in low-level image features, effective representation learning for the multi-level image features and distortion information is deemed crucial for the generalization of the BIQA method. Although this belief has inspired the core idea design of hierarchical feature fusion in numerous BIQA methodologies (such as MUSIQ (Ke et al., 2021), Hyper-IQA (Su et al., 2020) and Stair-IQA (Sun et al., 2022), etc.), the theoretical foundations underpinning this belief remain vacant and elusive, leading to poor interpretability of the success of these methods. Moreover, existing theoretical results are predominantly focused on classification tasks and fail to offer convincing support in the IQA domain. Therefore, in this work, we investigate the role of multi-level image features in the generalization ability of the BIQA framework in a fully-supervised regressive setting.

To conduct a rigorous theoretical analysis from the perspective of multi-level image features, we base our investigation on an unembellished CNN-based BIQA model, setting aside the complex structural designs or additional knowledge injection considered in some current BIQA models. Specifically, central to this work, three novel theorems on BIQA generalization and quality perception ability are presented. **For the role of low-level features**, under distribution invariance in training and test sets, we innovatively derive an upper bound for the Rademacher Average (RA)-based capacity term (Kakade et al., 2008) of the BIQA model, then we derive the corresponding generalization bound for the CNN-based BIQA framework. The theoretical results are summarized in Theorem 1, which indicates that the generalization ability tends to be reduced as the level of quality features increases, demonstrating the value of low-level image features for quality perception. In addition, under the challenging scenario where a distribution shift occurs in training and test sets, we prove a much tighter generalization bound than that in Theorem 1. The theoretical results are presented in Theorem 2, which builds on the conclusions of Theorem 1 to further elucidate the theoretical impact of distributional differences on generalization performance. **For the role of high-level features**, we prove that the BIQA networks tend to possess lower empirical errors and higher Betti number-based complexity (Zell, 1999) if they focus on learning higher-level quality features. The Betti number serves as a valuable measure of the representation power of BIQA models, and the theoretical results are presented in Theorem 3, which suggests that the high-level image features exert a positive impact on the quality perception capacity of BIQA networks, facilitating the improved robustness.

Based on the theoretical findings in the three proposed Theorems, we provide theoretical insights into why existing state-of-the-art IQA methods can effectively work with enhanced generalization. Additionally, there exist other valuable implications that can be further explored in these Theorems. For example, these theoretical findings reveal an inherent conflict between achieving robust generalization and strong representation power in BIQA networks, since the generalization ability and representation power tend to be reduced and increase respectively as the level of image features increases. On this basis, we offer some exemplary suggestions for developing BIQA systems that aim to effectively reduce empirical errors without compromising the generalization ability. Through cross-dataset evaluation experiments, these suggestions effectively address the tradeoff between generalization and representation, empirically validating our theoretical results. The key contributions of this study are summarized as follows:

- From a theoretical perspective, this work innovatively investigates the role of multi-level image features in the generalization and quality perception ability of the CNN-based BIQA models, which provides the theoretical guarantees for generalization research in the BIQA field. To the best of our knowledge, this is the first work to explicitly present the theoretical generalization bounds for the IQA model in the quality assessment domain.

- Under the conditions of distribution invariance or shift between training and test datasets, we respectively propose the different generalization bounds for CNN-based BIQA networks, with rigorous proofs emphasizing the crucial role of low-level features in generalization.

- Through Betti number-based analysis, we prove that BIQA networks focused on learning higher-level features tend to exhibit lower empirical errors and stronger representation abilities in quality perception, which validates the importance of high-level features.

- We uncover the fundamental conflict between the generalization and representation abilities of CNN-based BIQA models. Correspondingly, the proposed Theorems offer theoretically valid suggestions for BIQA training. Experimental results demonstrate the effectiveness of these suggestions, reflecting the reliability and practical value of our theoretical findings. In addition, the theoretical results in the proposed theorems can provide a global theoretical explanation for the good generalization of existing BIQA methods with varying designs.

## 2 RELATED WORKS

### 2.1 BLIND IMAGE QUALITY ASSESSMENT (BIQA)

BIQA has gained significant attention recently due to the absence of reference images in realistically distorted image datasets (Zhai & Min, 2020). With the development and wide applications of Deep Learning (DL), there spring up various DL-based methods achieving remarkable progress in BIQA (Ke et al., 2021; Golestaneh et al., 2022), such as RankIQA (Liu et al., 2017), CONTRIQUE (Madhusudana et al., 2022), GraphIQA (Simeng et al., 2023) and so on. Although these studies can improve the ability of quality perception on training datasets, their generalization is limited by the small scale of the training IQA datasets. To tackle this issue, some works try to improve the generalization of BIQA by more complex modules (Lin et al., 2020; Ma et al., 2017; Zhou et al., 2022) or unsupervised pre-training strategies (Prabhakaran & Swamy, 2023; Saha et al., 2023).In order to address different distributions between training and test sets, one of the latest methods is to integrate domain adaptive and ensemble learning into the IOA task (Roy et al., 2023). In the latest progresses in IQA field, some new studies are proposed to improve the generalization or robustness of IQA models by combining BIQA with various learning paradigms adapted to specific scenarios (Zhang et al., 2024; Wang et al., 2023; Yang et al., 2024; Zhang et al., 2022),such as Contrastive Learning, Continual Learning, Active Learning, Curriculum Learning, Multi-task Learning and Adversarial Learning. Although these existing IQA methods have achieved advanced performances in different settings and scenarios, the improvements in generalization in these existing methods come at the expense of training costs (Zhang et al., 2024; Zhong et al., 2024). Moreover, while effective representation learning for multi-level image features and distortion information is widely regarded as vital for the generalization of IQA methods (Ke et al., 2021; Su et al., 2020; Sun et al., 2022), theoretical guarantees for this belief remain elusive. Currently, there are no intuitive theoretical results addressing the generalization ability of BIQA models in existing literature. Consequently, discussions on the generalization of BIQA models are gaining traction. To our knowledge, this is the first work to explicitly present such generalization bounds in the IQA domain.

### 2.2 GENERALIZATION BOUND IN DEEP LEARNING

The most popular classification objectives in deep learning (such as cross-entropy loss) always encourage a larger output margin—the gap between the predicted true label and the next most confident label. These concepts predate deep learning and are backed by strong statistical guarantees for linear and kernel methods (Bartlett & Mendelson, 2002; Koltchinskii & Panchenko, 2002; Hofmann et al., 2008), which help explain the success of algorithms like SVM (Boser et al., 1992; Cortes, 1995). Nevertheless, deep learning always reveals statistical patterns that defy conventional understanding (Zhang et al., 2021; Neyshabur et al., 2017), offering fresh angles to investigate its generalization capabilities. These include insights into implicit and algorithmic regularization mechanisms (Soudry et al., 2018; Li et al., 2018), contemporary examinations of interpolation-based classifiers (Hastie et al., 2022; Bartlett et al., 2020), as well as examinations of the noise dynamics and stability of stochastic gradient descent (SGD) (Keskar et al., 2016; Chaudhari et al., 2019). More recently, the generalization boundary of Multilayer Perceptron (MLP) has been established in the algorithm selection tasks (Wu et al., 2024b;a). However, most of them are merely discussions focused on fully-connected neural networks. Although some works (Zhou & Feng, 2018; Long & Sedghi, 2019) have delved into the generalization of CNNs in recent years, they are only applicable to classification tasks, not regression tasks. In contrast, we innovatively introduce a generalization bound for regression networks in IQA tasks. Our theoretical analysis fully accounts for the unique characteristics of IQA tasks, effectively bridging the theoretical gap in the IQA field. We further discuss the applicability of our theoretical contributions to other regression tasks in Appedix H.

## 3 PRELIMINARIES

In the fully-supervised BIQA tasks, we assume $\mathcal{X} \subseteq \mathbb{R}^{H \times W \times 3}$ as the input space of distorted images, $\mathcal{Y} \subseteq [a, b]$ as the output space of MOS labels. $P$ is the joint distribution over $\mathcal{X} \times \mathcal{Y}$. We denote $S = \{(x_1, y_1), \cdots, (x_n, y_n)\}$ as the training dataset, each sample of which is i.i.d. sampled from $\mathcal{X} \times \mathcal{Y}$ based on the distribution $P$. And the goal is to train an effective MOS prediction model on $S$, i.e., $f \in \mathcal{F} : \mathcal{X} \to \mathbb{R}$ with the regression loss such as $L_1$ loss. Therefore, the expected error of the BIQA model $f$ on the test set can be measured by:

$$\text{err}_P(f) = \text{E}_{(x,y) \sim P} |f(x) - y| \tag{1}$$

We denote the loss in Eq. (1) evaluated on the training set $S$ as the training error (also referred to as the empirical error). The expected error on the test set is termed the test error, which serves as a proxy for assessing the prediction accuracy, as the true expected error cannot be directly obtained due to the unknown underlying data distribution $P$.

Without loss of generality, we consider a standard CNN-based BIQA network with $L$ layers following a similar architecture as described in (Sun et al., 2016). This includes $L - 1$ hidden layers for quality perception feature extraction and an output layer for MOS prediction. We assume that there are $m_l$ units in the $l$-th layer ($l = 1, \cdots, L$), with $m_L = 1$. To mitigate overfitting, common practice is to impose constraints on the magnitude of the weights, for instance, by introducing a constraint $A$ on the summation of weights for each unit. This can potentially enhance the generalization capability of the model on unseen datasets. Mathematically, we denote the function space of multi-layer neural networks with depth $L$ and weight constraint $A$ as $\mathcal{F}_A^L$:

$$\mathcal{F}_A^L = \left\{ x \to \sum_{i=1}^{m_{L-1}} w_i f_i(x); f_i \in \mathcal{F}_A^{L-1}, \sum_{i=1}^{m_{L-1}} |w_i| \leq A, w_i \in \mathbb{R} \right\} \tag{2}$$

Similarly, the $l$-th hidden layer ($l = 1, \cdots, L - 1$) can be denoted as:

$$\mathcal{F}_A^l = \left\{ x \to \varphi \left( \phi \left( f_1(x) \right), \cdots, \phi \left( f_{p_l}(x) \right) \right) \mid f_1, \cdots, f_{p_l} \in \overline{\mathcal{F}}_A^l \right\} \tag{3}$$

where $\overline{\mathcal{F}}_A^l$ is calculated as:

$$\overline{\mathcal{F}}_A^l = \left\{ x \to \sum_{i=1}^{m_{l-1}} w_i f_i(x); f_i \in \mathcal{F}_A^{l-1}, \sum_{i=1}^{m_{l-1}} |w_i| \leq A, w_i \in \mathbb{R} \right\} \tag{4}$$

and we denote $\mathcal{F}_A^0$ as:

$$\mathcal{F}_A^0 = \{x \to x_{i,j,k}; i \in \{1, \cdots, H\}; j \in \{1, \cdots, W\}; k \in \{1, 2, 3\}\} \tag{5}$$

The functions $\varphi$ and $\phi$ are the pooling function and activation function respectively. $x_{i,j,k}$ denotes the pixel value in image $x$, and $w_i$ is the weight parameter in the BIQA network.

Note that Eq. (3), representing the hidden layer, serves as a unified representation encompassing both convolutional and fully connected layers. Specifically: **(i)** For a fully connected layer at the $l$-th level, $p_l = 1$ and $\varphi(x) = x$. This implies that the $(l - 1)$-th layer's outputs undergo linear combination and subsequent activation to transition to the $l$-th layer. **(ii)** In the case of a convolutional layer at the $l$-th level, the transition from the outputs of the $(l - 1)$-th layer to the $l$-th layer involves filtering, activation, and pooling. Consequently, numerous weight parameters in Eq. (4) become zero, and $m_l$ is determined by $m_{l-1}$ along with the number and size of the convolution kernels.

In Eq. (3), $p_l$ represents the size of the pooling region in the $l$-th layer. Commonly used pooling functions $\varphi : \mathbb{R}^{p_l} \to \mathbb{R}$ include max-pooling $\max(t_1, \cdots, t_{p_l})$ and average-pooling $(t_1 + \cdots + t_{p_l})/p_l$. The frequently employed activation functions are typically 1-Lipschitz, such as the sigmoid function $\phi(t) = \frac{1}{1+e^{-t}}$, the hyperbolic tangent function $\phi(t) = \frac{e^t - e^{-t}}{e^t + e^{-t}}$, and the rectifier function $\phi(t) = \max(0, t)$. During training the BIQA models, the back-propagation method is typically utilized to minimize loss functions in Eq (1) on the training set, where the weight parameters undergo updates through the application of stochastic gradient descent (SGD).

## 4 THE ROLE OF LOW-LEVEL IMAGE FEATURES IN IQA

In this section, we present the expected errors and generalization error bounds for the aforementioned standard BIQA network models under distribution invariance and distribution shift respectively, which can both reveal the key role of low-level image features in IQA.

## 4.1 EXPECTED ERROR OF BIQA MODELS

To theoretically expose the effects of different architecture parameters of BIQA model on the generalization performance, we first prove that the setting of $L_1$ loss in BIQA tasks satisfies the Lipschitz condition (Valentine, 1945). This is a necessary prerequisite for deriving expected errors and establishing the generalization bound for BIQA tasks, which we subsequently introduce.

**Lemma 1.** *The $L_1$ loss function satisfies the Lipschitz condition with Lipschitz constant $L_\ell = 1$.*

*Proof.* Please refer to the Appendix A.    □

Now we present the upper bound of the expected error for BIQA models, which provides an intuitive reflection of their generalization capability. Similar to the expected errors in Eq.(1), we can define the empirical error as $\text{err}_S(f) = \text{E}_{(x,y) \in S} |f(x) - y|$. Building upon Lemma 1, we can derive the generalization bound (i.e. $\text{err}_P(f) - \text{err}_S(f)$) for BIQA models trained with the $L_1$ loss:

**Lemma 2** ((Bartlett & Mendelson, 2002)). *Assume the loss $\ell$ is Lipschitz (with respect to its first argument) with Lipschitz constant $L_\ell$ and that $\ell$ is bounded by $c$. For any $\delta > 0$ and with probability at least $1 - \delta$ simultaneously for all $f \in \mathcal{F}$, we have the upper bound of the expected error:*

$$\text{err}_P(f) \le \text{err}_S(f) + 2L_\ell \mathcal{R}_n(\mathcal{F}) + c\sqrt{\frac{\log(1/\delta)}{2n}} \tag{6}$$

*where $n$ is the sample size. $\mathcal{R}_n(\mathcal{F})$ is the Rademacher complexity (Kakade et al., 2008) of a function class $\mathcal{F}$, details of which are shown in Definition 3 in the Appendix B.*

According to the upper bound presented in Lemma 2, the expected error of a BIQA model can be upper bounded by the sum of three terms: the complexity term RA, the empirical error, and a term related to the sample size. In the following two subsections, we will delve into the generalization bounds for BIQA models under distribution invariance and distribution shift respectively.

## 4.2 GENERALIZATION BOUNDS FOR BIQA MODELS UNDER DISTRIBUTION INVARIANCE

In this subsection, we study the role of quality feature level in the RA-based capacity term with training and testing data distributions aligned. We first derive a uniform upper bound of RA for CNN-based BIQA networks, based on which the generalization bound for BIQA models is derived.

**Theorem 1.** *Assume the input space $\mathcal{X} = [0, M]^{H \times W \times 3}$. In the deep BIQA neural networks, if activation function $\phi$ is non-negative and satisfies the Lipschitz condition with Lipschitz constant $L_\phi$, pooling function $\varphi$ is max-pooling or average-pooling, and the size of pooling region in each layer is bounded, i.e., $p_l \le p$, then we have:*

$$R_n\left(\mathcal{F}_A^L\right) \le bMA\sqrt{\frac{\ln(3HW)}{n}} (pL_\phi A)^{L-1} \tag{7}$$

*where $b$ is a constant. Specially, if $pL_\phi A > 1$, then for any $\delta > 0$ and with probability at least $1 - \delta$ simultaneously for all $f \in \mathcal{F}_A^L$ with 1-Lipschitz positive-homogeneous activation functions, we have:*

$$\text{err}_P(f) - \text{err}_S(f) \le \mathcal{O}\left(\frac{(pL_\phi A)^L}{n^{\frac{1}{2}}}\right) + c\sqrt{\frac{\log(1/\delta)}{2n}} \tag{8}$$

*where $c$ is the upper bound of $L_1$ loss.*

*Proof.* Please refer to the Appendix B.    □

Theorem 1 provides a theoretical guarantee for the generalization capacity of standard BIQA networks. The three weak assumptions in Theorem 1 - namely, (1) $p_l \le p$, (2) the pixel values satisfy $0 \le x_{i,j,k} \le M$, and (3) the activation function is non-negative and $L_\phi$-Lipschitz - can be readily satisfied in the majority of practical scenarios. These assumptions do not impose significant limitations on the scope of the theorem but rather serve to ensure the theoretical correctness of the results.

From the theoretical result in Eq. (8) of Theorem 1, we can make the following key observations and conclusions: **(1)** With an increasing depth $L$, the upper bound of the RA term will increase, leading

to a looser generalization bound. This indicates that relying solely on high-level quality perception features can have a negative impact on the generalization ability of BIQA models. This highlights the importance of low-level image features for the generalization of IQA. **(2)** As the sample size $n$ increases, the generalization bound will decrease, which aligns with the common understanding of neural network training. This also theoretically confirms that insufficient training data is a major reason behind the poor generalization ability of existing BIQA models. **(3)** Smaller size of the pooling region $p$ and a tighter weight parameter boundary $A$ may promote better generalization. This motivates us to apply a regularization penalty on the model parameters during the training process.

### 4.3 TIGHTER GENERALIZATION BOUND OF BIQA MODELS UNDER DISTRIBUTION SHIFT

In Theorem 1, the generalization bound is not tight enough since it exponentially grows with depth $L$. In addition, the theoretical result in Eq. (8) ignores the effect of distribution difference on the generalization of BIQA models. Nevertheless, the effect of the distribution shift from training set to test set on generalization performance is significant, leading to a series of domain adaptation efforts on the IQA domain. Therefore, it is meaningful to propose a tighter generalization boundary related to the distribution difference between training set and test set for BIQA models.

Similar to Theorem 1, we denote the space of BIQA models as $\mathcal{F}$ in Section 3, where $f(x) \in \mathcal{F}$ maps the image sample to real-valued MOS label. Since the convolutional layer and fully-connected layer can be uniformly represented by Eq. (3) and Eq. (4), and the functionality of CNN can also be implemented by MLP (Tolstikhin et al., 2021), for analytical convenience, we consider BIQA network $f$ as a unified MLP, including all sub-MLPs in different levels of quality perception representations. This transformation is mathematically equivalent, as all parameters and activation functions in this unified MLP can be derived from the original model without extra computation (Wu et al., 2024c). Based on Eq. (23) in Definition 3 in the Appendix B and the above assumptions, we derive the Rademacher Complexity of BIQA models from a new perspective:

**Lemma 3.** *Let $n$ denote the number of image samples in the training set. $W_i$ represents the parameter matrix in the $i$-th layer, and $L$ denote the number of layers. Then, for the BIQA model class $\mathcal{F}_A^L$, the Rademacher complexity of BIQA model is bounded by:*

$$R_{n,\boldsymbol{\eta}}\left(\mathcal{F}_A^L\right) \le \frac{1}{n\lambda} \log\left(2^L \cdot \mathbb{E}_{\boldsymbol{\epsilon}} \exp\left(M\lambda Q\right)\right) = \frac{L\log 2}{n\lambda} + \frac{1}{n\lambda} \log\left(\cdot \mathbb{E}_{\boldsymbol{\epsilon}} \exp\left(M\lambda Q\right)\right) \quad (9)$$

*where $x_i$ denotes the $i$-th instance, $\epsilon_i$ is a Rademacher variable, $\lambda$ is a random variable, and*

$$M = \prod_{j=1}^{L} M_F(j), \quad Q = \left\| \sum_{i=1}^{n} \epsilon_i \eta_i x_i \right\| \quad (10)$$

*where $M_F(j)$ denotes the upper-bound of $\|W_j\|_F$, and $\eta_i = P_{test}(x_i)/P_{train}(x_i)$,*

*Proof.* Please refer to the Appendix C. □

According to Lemma 3, the generalization bound with RA of Eq. (9) is tighter than that in Theorem 1 by getting rid of the exponential dependence of $L$, and note that Eq. (9) holds for each $\lambda \in \mathbb{R}$, based on which we can discuss the generalization performance of BIQA model in Theorem 2.

**Theorem 2.** *Follow the notation in Lemma 3, and let $D\left(P_{test}\|P_{train}\right)$ denote the chi-squre divergence between training and test distribution, $L_\ell$ denote the Lipschitz constant of loss function $\ell$, and $L$ denote the number of layers. Then, for the BIQA model class $\mathcal{F}_A^L$, with probability at least $1 - \delta$ simultaneously for all $f \in \mathcal{F}_A^L$, we have:*

$$\mathrm{err}_P(f) \le \mathrm{err}_S(f) + \mathcal{O}\left(\frac{L_\ell \sqrt{LM} \cdot \sqrt{D\left(P_{test}\|P_{train}\right) + 1}}{\sqrt{n}}\right) + c\sqrt{\frac{\log(1/\delta)}{2n}} \quad (11)$$

*Proof.* Please refer to the Appendix D. □

From Eq. (11), we can conclude that: **(1)** Theorem 2 provides a theoretical guarantee that the greater the distribution difference, the worse the generalization performance. **(2)** The generalization boundary is linearly and positively correlated to the Lipschitz constant of the BIQA loss function, suggesting that the choice of loss function has a significant influence on the generalization capability of the BIQA model. **(3)** The greater the value of $M = \prod_{i=1}^{L} M_F(i)$ in Eq. (10), the larger the upper bound, which implies that an excessively large number of parameters may diminish the generalization ability of the BIQA model, corroborating the Conclusion **(3)** drawn in Theorem 1. **(4)** Same as Conclusion **(1)** from Theorem 1, learning low-level image features is crucial for the generalization of IQA models.

# 5 THE ROLE OF HIGH-LEVEL IMAGE FEATURES IN IQA

To reveal the role of high-level image features, we study the role of the quality feature representation level in shaping the empirical error term, as this reflects the representation power of BIQA models.

## 5.1 BETTI NUMBERS

We utilize the Betti numbers (Bianchini & Scarselli, 2014) based complexity to measure the representation power of the CNN-based BIQA networks. For this purpose, We first present our definition of the Betti numbers-based complexity for BIQA models, which can be proved to be reasonable according to the formal definition of Betti numbers presented in Definition 4 in the Appendix E.1. Subsequently, we prove the existence of an upper bound on the Betti numbers-based complexity of BIQA models, followed by a discussion of the relevant analysis.

Inspired by (Sun et al., 2016), we can reasonably generalize the definition of Betti numbers-based complexity into regression setting for BIQA models as follows.

**Definition 1.** *The Betti numbers-based complexity of functions implemented by BIQA neural networks $\mathcal{F}_A^L$ is defined as $N(\mathcal{F}_A^L) = \sum_{i=1}^{K} B(S_i)$, where $B(S_i)$ is the sum of Betti numbers that measures the complexity of the set $S_i$. Here $S_i = \left\{ x \in \mathbb{R}^{H \times W \times 3} \mid a + \frac{b-a}{K}(i+1) \geq f(x) \geq a + \frac{b-a}{K}i \right\}$, where $i = 0, \ldots, K - 1$.*

As shown in Definition 1, the Betti numbers-based complexity takes into account the MOS output and merges the image samples with similar quality levels (thus is more accurate than the linear region number complexity (Montufar et al., 2014) in measuring the representation power). It is worth noting that the parameter $K$ determines the granularity of the division of quality levels, which can be dynamically adjusted. To the best of our knowledge, existing works have only derived bounds on the Betti numbers-based complexity in the scenario of classification tasks, and there are no such results for the regression setting or convolutional BIQA networks, leaving a theoretical gap.

## 5.2 REPRESENTATION POWER OF BIQA MODELS

In the subsection, we propose our own Theorem 3 to address the above-mentioned theoretical gap.

**Theorem 3.** *For BIQA networks $\mathcal{F}_A^L$ with $h$ hidden units, if activation function $\phi$ is a Pfaffian function with complexity $(\alpha, \beta, \eta)$, pooling function $\varphi$ is average-pooling and $3HW \leq h\eta$, then we have:*

$$N\left(\mathcal{F}_A^L\right) \leq K 2^{d+h\eta(h\eta-1)/2} \times \mathcal{O}\left((d(\xi(L-1) + \beta(\alpha+1)))^{d+h\eta}\right) \tag{12}$$

*where $\xi = \alpha + \beta - 1 + \alpha\beta$ and $d = 3HW$.*

*Proof.* Please refer to the Appendix E.2. □

Theorem 3 establishes an upper bound on the Betti numbers-based complexity for BIQA networks with general activation functions. For specific activation functions, we derive the following corollaries:

**Corollary 1.** *Let $d = 3HW$, if $\phi = \arctan(\cdot)$ and $3HW \leq 2h$, we have:*

$$N\left(\mathcal{F}_A^L\right) \leq K 2^{d+h(2h-1)} \times \mathcal{O}\left((d(L-1) + d)^{d+2h}\right) \tag{13}$$

*if $\phi = \tanh(\cdot)$ and $3HW \leq h$, we have:*

$$N\left(\mathcal{F}_A^L\right) \leq K 2^{d+h(h-1)/2} \times \mathcal{O}\left((d(L-1) + d)^{d+h}\right) \tag{14}$$

*Proof.* Please refer to the Appendix E.3. □

According to Corollary 1, we can observe that the representation power of BIQA models with activation function $\phi = \arctan(\cdot)$ tends to be stronger than that with activation function $\phi = \tanh(\cdot)$.

Through Theorem 3, we can make the following conclusions: **(1)** As the depth $L$ increases, the Betti numbers-based complexity grows. This indicates that learning high-level quality perception features has a positive impact on the representation power of BIQA models, allowing them to better fit the training data and achieve smaller empirical errors. Our experiments show that this theoretical result in Eq. (12) is consistent with our empirical observations on different datasets. **(2)** In BIQA networks, higher Pfaffian-based complexity of the activation function leads to stronger representation ability.

# 6 DISCUSSIONS

In this section, we first reveal the conflict of strong representation power and robust generalization in BIQA models based on the theoretical analysis in Section 4 and Section 5. Then the theoretical results in the three proposed theorems inspire us to provide a global theoretical explanation for existing BIQA methods with varying designs that achieve good generalization. On these bases, we offer some exemplary suggestions for enhancing the generalization in BIQA models.

## 6.1 CONFLICT OF STRONG REPRESENTATION AND GENERALIZATION IN BIQA MODELS

In addition to the conclusions in Sections 4 and 5, there exist some other valuable insights that can be explored in these proposed and proved Theorems. This subsection is an example.

Based on the discussions about the generalization bound and representation power of BIQA models in Section 4 and Section 5, we can observe that: as the level of quality perception feature increases, (1) the RA term and the generalization bound of BIQA model increases (as stated in Theorem 1 and Theorem 2); (2) the empirical error tends to decrease due to the greater representational power of BIQA networks (as stated in Theorem 3). Consequently, it can be concluded that for BIQA models with a limited number of hidden units, emphasis on learning only the high-level quality perception feature is not always beneficial, as there is a distinct trade-off between good generalization bound and representation power in BIQA models. Specifically, we have the following conclusion: As the level of quality perception feature increases, the test error of the BIQA networks may first decrease and then increase. This conclusion is proved in the Section 7 of Experiments.

## 6.2 THEORETICAL EXPLANATION FOR EXISTING IQA MODELS

In existing representative DL-based IQA models, various schemes are proposed to enhance the generalization, which can be roughly divided into three categories: training with (**1**) extra datasets or network branches (such as CONTRIQUE (Madhusudana et al., 2022) and LIQA (Zhang et al., 2022)); (**2**) superior loss function ( such as NIMA (Talebi & Milanfar, 2018) and Norm-in-Norm (Li et al., 2020)); (**3**) effective feature fusion, such as MUSIQ (Ke et al., 2021), Hyper-IQA (Su et al., 2020) and Stair-IQA (Sun et al., 2022), etc. The enhancing generalization in Category (**1**) is apparent and intuitive, which is similar to the enlarging data scale $n$ in Eqs. (8) and (11). For Category (**2**), since the Lipschitz constant $L_\ell$ of the loss function is strongly related to the generalization bound, as shown in Eq. (31) in the proof of Theorem 1 and Eq. (11) in Theorem 2, an appropriate loss function can facilitate better generalization. For Category (**3**), according to the analysis in Section 6.1, the roles of low-level image features and high-level image features are complementary, hence multi-level image quality feature learning and fusion are theoretically valid for generalization enhancement. Additionally, for BIQA models using test-time adaptation (TTA) (Roy et al., 2023), enhanced generalization can be theoretically explained by Theorem 2, as TTA models reduce distribution shifts between the training and testing sets by unsupervised fine-tuning on test samples before testing.

## 6.3 EXEMPLARY SUGGESTIONS RELATED TO PROPOSED THEOREMS

The proposed theorems can not only provide theoretical support for existing works, but also offer valuable insights for further exploration. Accordingly, we present some examples as practical suggestions for BIQA network design based on these theorems: (**1**) In the feature perspective, according to the Conclusion (**1**) from Theorem 1 and the Conclusion (**1**) from Theorem 3, the roles of low-level and high-level are complementary. Therefore, we propose fusing multi-level features to address distortion complexity, enhancing generalization ability. Figure 2 in Appendix G.2 exemplifies this approach by combining two levels of features. (**2**) In the loss perspective, based on Eq. (31) in the proof of Theorem 1 and Eq. (11) in Theorem 2, we propose to improve the loss function. For example, by incorporating a regularization term, which aims to minimize empirical error and enhance representation power without increasing network depth, the conflict between high-level and low-level image features can be avoided. (**3**) In the perspective of network parameters, according to the Conclusion (**3**) from Theorem 1 and Eq. (10) in Theorem 2, we propose to constrain the size of the network parameters with another regularization term. Details of theoretical explanations for these suggestions are provided in Appendix F. Based on the Suggestion (**2**) and Suggestion (**3**), we have

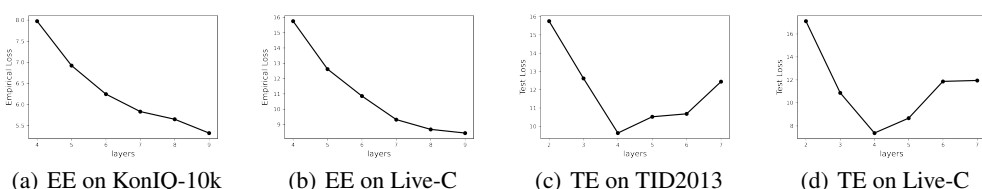

| (a) EE on KonIQ-10k | (b) EE on Live-C | (c) TE on TID2013 | (d) TE on Live-C |

Figure 1: (a-b): Impact of depth on empirical error (EE). (c-d): Impact of depth on test error (TE).

the exemplary loss function:

$$L_{BIQA} = L_1\left(f(x), y\right) + \mu Norm\left\{\cos\left[R_B\left(x_f\right), R_B\left(y\right)\right]\right\} + \nu\|\mathbf{W}_L\|_F \tag{15}$$

where $x_f$ denotes the extracted quality perception feature vector, and $R_B\left(x_f\right)$ represents the order (from smallest to largest) of Euclidean distances between $x_f$ and the feature vectors of other image samples in the same batch $B$. cos and $Norm$ mean the cosine similarity and max-min normalization with maximum value 1 and the minimum value $\cos([1, 2, ..., B-1], [B-1, B-2, ..., 1])$. The second term (consistency regularizer) in Eq. (15) corresponds to Suggestion (**2**), which can directly minimize empirical error by remaining the consistency between feature space and label space. The third term (parameter regularizer) in Eq. (15) corresponds to Suggestion (**3**), where $\mathbf{W}_L$ denotes the weight parameter of the final prediction layer, and $\|\cdot\|_F$ denotes the Frobenius norm. $\mu$ and $\nu$ are the hyper-parameters. More detailed descriptions about Eq. (15) are shown in Appendix G.3.

# 7 EXPERIMENTS

We empirically validate the theoretical results presented earlier through extensive experiments. Details about the datasets and experimental settings used in this study are provided in the Appendix G.1.

**Experimental Verification of Theorem 3** According to Theorem 3, we have concluded that: With the increase of $L$ and the learning level of quality perception features, the empirical error of the BIQA model will decrease. To investigate this, we train CNN based on the official demo with different depths $L$ and restricted number of hidden units on KonIQ-10k (Hosu et al., 2020) and Live-C (Ghadiyaram & Bovik, 2015), and the quality perception features are extracted in the $(L-1)$-th layer. The experimental results are shown in (a-b) of Figure 1 and indicate that deeper BIQA networks learning high-level quality perception features have smaller empirical errors than shallower BIQA networks learning low-level quality features, which can verify our Theorem 3.

**Experimental Verification of Theorem 1 and Discussion Results of Section 6.1** According to Theorem 1, as the depth L increases, the generalization ability of the BIQA model diminishes. Therefore, we propose the conjecture in Section 6.1: As the level of quality perception feature increases, the test error and of the BIQA networks may first decrease and then increase. To investigate this, we train the same CNN demo as above with different depths $L$ and restricted number of hidden units on KonIQ-10k (Hosu et al., 2020), and then test on TID2013 (Ponomarenko et al., 2015) and Live-C (Ghadiyaram & Bovik, 2015) respectively. The experimental results are shown in (c-d) of Figure 1, which can verify the conclusion of Theorem 1 and Section 6.1.

**Impacts of Different Hyper-parameters in Eq. (15)** According to the theoretical results of this work, we propose the Consistency Regularizer and Parameter Regularizer in the loss function Eq. (15). For Parameter Regularizer, it is worth mentioning that we only consider the vector of weight parameter since there are numerous layers in a deep BIQA network, leading to too many hyperparameters, and careful adjustments for them are not practical. We train the BIQA network with the backbone of ResNet-18 (He et al., 2016) on KonIQ-10k (Hosu et al., 2020) and then test on CID2013 (Virtanen et al., 2014). The PLCC and SRCC results are recorded on Tables 1 and 2 with different $\mu$ and $\nu$.

**Ablation Study for Suggestions in Section 6.3** According to the theoretical results and related analysis, we propose three exemplary and universal suggestions for the design of BIQA models, which serve as proof of the practical value of the proposed theorems. We denote $B$, $S_1$, $S_2$ and $S_3$ as the baseline of Resnet50 (He et al., 2016) (same as the example of Figure 2 in Appendix G.2), Suggestion (**1**), Suggestion (**2**) and Suggestion (**3**), respectively, where $\mu = 10$ and $\nu = 0.01$. The experimental results of which are summarized in Table 3 and Table 4, which proves the rationality

Table 1: The impacts of $\mu$ with fixed $\nu = 0.01$. BIQA model (ResNet-18 (He et al., 2016)) is trained on KonIQ-10k (Hosu et al., 2020) and tested on CID2013 (Virtanen et al., 2014).

| $\mu$ | 0 | 1 | 5 | 10 | 15 |
|---|---|---|---|---|---|
| PLCC | 0.681 | 0.685 | 0.702 | **0.713** | 0.708 |
| SRCC | 0.685 | 0.679 | 0.692 | **0.697** | 0.684 |
| RMSE | 13.37 | 14.16 | 13.05 | **12.21** | 12.74 |

Table 2: The impacts of $\nu$ with fixed $\mu = 10$. BIQA model (ResNet-18 (He et al., 2016)) is trained on KonIQ-10k (Hosu et al., 2020) and tested on CID2013 (Virtanen et al., 2014).

| $\nu$ | 0 | 0.01 | 0.05 | 0.1 | 1 |
|---|---|---|---|---|---|
| PLCC | 0.709 | **0.713** | 0.712 | 0.704 | 0.690 |
| SRCC | 0.687 | 0.697 | **0.701** | 0.695 | 0.678 |
| RMSE | 13.19 | 12.21 | **12.06** | 12.89 | 13.52 |

Table 3: Ablation Study on KonIQ-10k (Hosu et al., 2020), it is divided into 8:2 for training and testing.

| Models | PLCC | SRCC | RMSE |
|---|---|---|---|
| $B$ | 0.857 | 0.865 | 7.005 |
| $B+S_1$ | 0.863 | 0.872 | 6.846 |
| $B+S_1+S_2$ | 0.872 | 0.886 | 6.703 |
| $B+S_1+S_2+S_3$ | **0.873** | **0.892** | **6.674** |

Table 4: Cross-data Ablation Study on KADID-10k (Virtanen et al., 2014) (train) and CID2013 (Virtanen et al., 2014) (test).

| Models | PLCC | SRCC | RMSE |
|---|---|---|---|
| $B$ | 0.711 | 0.689 | 12.91 |
| $B+S_1$ | 0.718 | 0.705 | 12.27 |
| $B+S_1+S_2$ | **0.725** | 0.726 | **11.68** |
| $B+S_1+S_2+S_3$ | 0.719 | **0.731** | 11.80 |

of the suggestions and theoretical results. The fact that the effectiveness of Suggestion (**3**) is not significant may be due to the consideration of only the weights of the prediction layer, neglecting the other weight parameters in the network. The reasons for this have been explained above.

**Experimental Verification of Theorem 2**   We train the BIQA network with backbone of ResNet-18 on KonIQ-10k, which is divided to two subsets with different distributions. Specifically, we divide the dataset into low-score set $LS$ and high-score set $HS$ based on the median of their MOS labels. Then $LS$ and $HS$ are then divided into 1:9 respectively, termed as $LS_1$, $LS_9$ and $HS_1$, $HS_9$. Subsequently, $LS_9$ and $HS_1$ are combined as $F_1$, $LS_1$ and $HS_9$ are combined as $F_2$, hence the distributions of $F_1$ and $F_2$ are different. In order to simulate various distribution differences between different training sets and test sets, we divided $F_1$ into 2 equal parts named $F_1^1$ and $F_1^2$ randomly, where $F_1^1$ is used as the training set, and $F_1^2$ and $F_2$ are combined as test set. For the convenience of presentation, $F_1^{\text{test}}:F_2^{\text{test}}$ denotes the proportion of the two distributions $F_1$ and $F_2$ in the test set, and $F_1^{\text{train}}$ denotes the training set from distribution $F_1$. The whole construction process can be referred intuitively in Figure 4 in the Appendix G.1. According to Table 9, we can observe that the greater the distribution difference, the worse the generalization performance.

Table 5: The impact of distribution differences on generalization performances of BIQA model. $F_1^{\text{test}}:F_2^{\text{test}}$ denotes the proportion of the two different distributions in test set. $F_1^{\text{train}}$ denotes the training set from $F_1$. The experiments are conducted on dataset KonIQ-10k (Hosu et al., 2020).

| $F_1^{\text{train}}:(F_1^{\text{test}}:F_2^{\text{test}})$ | 1:(1:0) | 1:(1:0.5) | 1:(1:1) | 1:(1:2) | 1:(0:2) |
|---|---|---|---|---|---|
| PLCC | **0.818** | 0.775 | 0.746 | 0.723 | 0.695 |
| SRCC | **0.807** | 0.784 | 0.759 | 0.738 | 0.716 |
| RMSE | **8.308** | 8.621 | 9.248 | 9.874 | 10.122 |

# 8   CONCLUSION

This paper innovatively investigates the role of multi-level image features in the generalization and quality perception ability of the CNN-based BIQA models from a theoretical perspective. Specifically, we first propose and rigorously prove the generalization bounds for CNN-based BIQA networks under the conditions of distribution invariance or shift between training and test datasets, which prove the crucial role of low-level features. Then we validate the importance of high-level features through Betti number-based analysis with rigorous mathematical proofs. Based on the theoretical results, we can provide theoretical explanations for the enhanced generalization of existing methods. Furthermore, we uncover the inherent conflict between the generalization capacity and representation power of IQA models. Correspondingly, we can offer theoretically valid suggestions for BIQA training based on our Theorems. To our knowledge, this is the first work in the IQA domain to establish a theoretical generalization boundary, thereby filling an important gap in the theoretical understanding of the generalization of IQA models.

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

## A   THE PROOF OF LEMMA 1

We first give the definition of Lipschitz condition.

**Definition 2** ((Valentine, 1945))**.** *A loss function $\ell$ is Lipschitz (with respect to its first argument) if there exists a constant $L > 0$ such that, for any $x_1, x_2$ (belonging to the domain of the first argument of $\ell$) and any fixed value $y$ (for the other arguments of $\ell$, if any), the following inequality holds:*

$$|\ell(x_1, y) - \ell(x_2, y)| \leq L|x_1 - x_2| \tag{16}$$

*where $L$ is named as Lipschitz constant.*

To prove that the $L_1$ loss function satisfies the Lipschitz condition, let $\mathbf{x}, \mathbf{y}, \mathbf{z} \in \mathbb{R}^n$ be arbitrary vectors. The $L_1$ loss function between $\mathbf{x}$ and $\mathbf{y}$ is defined as:

$$L_1(\mathbf{x}, \mathbf{y}) = \sum_{i=1}^{n} |x_i - y_i|. \tag{17}$$

To show that $L_1$ satisfies the Lipschitz condition, we need to find a constant $K \geq 0$ such that

$$|L_1(\mathbf{x}, \mathbf{y}) - L_1(\mathbf{x}, \mathbf{z})| \leq K \cdot \|\mathbf{y} - \mathbf{z}\|_1, \tag{18}$$

where $\|\mathbf{y} - \mathbf{z}\|_1 = \sum_{i=1}^{n} |y_i - z_i|$ is the $L_1$ norm. Consider the absolute difference in $L_1$ loss:

$$|L_1(\mathbf{x}, \mathbf{y}) - L_1(\mathbf{x}, \mathbf{z})| = \left| \sum_{i=1}^{n} |x_i - y_i| - \sum_{i=1}^{n} |x_i - z_i| \right|. \tag{19}$$

By the triangle inequality for absolute values, we have

$$|a - b| \leq |a| + |b|, \tag{20}$$

which implies

$$||x_i - y_i| - |x_i - z_i|| \leq |(x_i - y_i) - (x_i - z_i)| = |y_i - z_i|. \tag{21}$$

Summing over all $i$, we obtain

$$|L_1(\mathbf{x}, \mathbf{y}) - L_1(\mathbf{x}, \mathbf{z})| \leq \sum_{i=1}^{n} |y_i - z_i| = \|\mathbf{y} - \mathbf{z}\|_1. \tag{22}$$

Therefore, the $L_1$ loss satisfies the Lipschitz condition with $K = 1$.

## B   THE PROOF OF THEOREM 1

We first give the definition of Rademacher Average.

**Definition 3** ((Kakade et al., 2008))**.** *Suppose $\mathcal{F} : \mathcal{X} \to \mathbb{R}$ is a model space with a single dimensional output. The Rademacher Average (RA) (also known as Rademacher Complexity) of $\mathcal{F}$ is defined as follows:*

$$\mathcal{R}_n(\mathcal{F}) = \mathbb{E}_{\mathbf{x}, \sigma} \left[ \sup_{f \in \mathcal{F}} \frac{1}{n} \sum_{i=1}^{n} f(x_i) \sigma_i \right] \tag{23}$$

*where $\sigma_i$ independently takes values in $\{+1, -1\}$ with equal probability. $\mathbf{x} = \{x_1, \cdots, x_n\} \sim P_x^n$.*

According to the definition of $\mathcal{F}_A^L$ and RA, we have:

$$
\begin{aligned}
R_n\left(\mathcal{F}_A^L\right) &= \mathbf{E}_{\mathbf{x}, \boldsymbol{\sigma}} \left[ \sup_{\|\mathbf{w}\|_1 \leq A, f_j \in \mathcal{F}_A^{L-1}} \left| \frac{2}{n} \sum_{i=1}^{n} \sigma_i \sum_{j=1}^{m_{L-1}} w_j f_j(x_i) \right| \right] \\
&= \mathbf{E}_{\mathbf{x}, \boldsymbol{\sigma}} \left[ \sup_{\|\mathbf{w}\|_1 \leq A, f_j \in \mathcal{F}_A^{L-1}} \left| \frac{2}{n} \sum_{j=1}^{m_{L-1}} w_j \sum_{i=1}^{n} \sigma_i f_j(x_i) \right| \right]
\end{aligned}
\tag{24}
$$

Supposing $\mathbf{w} = \left\{ w_1, \cdots, w_{m_{L-1}} \right\}$ and $\mathbf{h} = \left\{ \sum_{i=1}^{n} \sigma_i f_1(x_i), \cdots, \sum_{i=1}^{n} \sigma_i f_{m_{L-1}}(x_i) \right\}$, the inner product $\langle \mathbf{w}, \mathbf{h} \rangle$ is maximized when $\mathbf{w}$ is at one of the extreme points of the $l_1$ ball, which implies:

$$R_n\left(\mathcal{F}_A^L\right) \leq A \mathbf{E}_{\mathbf{x}, \boldsymbol{\sigma}} \left[ \sup_{f \in \mathcal{F}_A^{L-1}} \left| \frac{2}{n} \sum_{i=1}^{n} \sigma_i f(x_i) \right| \right] \tag{25}$$

$$= A R_n\left(\mathcal{F}_A^{L-1}\right).$$

For function class $\mathcal{F}_A^{L-1}$, if the $(L-1)$-th layer is a fully connected layer, it is clear that $R_n\left(\mathcal{F}_A^{L-1}\right) \leq R_n\left(\phi \circ \overline{\mathcal{F}}_A^{L-1}\right)$ holds. If the $(L-1)$-th layer is a convolutional layer with max-pooling or average-pooling, we have,

$$R_n\left(\mathcal{F}_A^{L-1}\right)$$

$$= \mathbf{E}_{\mathbf{x}, \boldsymbol{\sigma}} \left[ \sup_{f \in \mathcal{F}_A^{L-1}} \left| \frac{2}{n} \sum_{i=1}^{n} \sigma_i f(x_i) \right| \right]$$

$$\leq \mathbf{E}_{\mathbf{x}, \boldsymbol{\sigma}} \left[ \sup_{f_1, \cdots, f_{p_{L-1}} \in \overline{\mathcal{F}}_A^{L-1}} \left| \frac{2}{n} \sum_{i=1}^{n} \sigma_i \sum_{j=1}^{p_{L-1}} \phi\left(f_j(x_i)\right) \right| \right] \tag{26}$$

$$= \mathbf{E}_{\mathbf{x}, \boldsymbol{\sigma}} \left[ \sup_{f_1, \cdots, f_{p_{L-1}} \in \overline{\mathcal{F}}_A^{L-1}} \left| \frac{2}{n} \sum_{j=1}^{p_{L-1}} 1 \sum_{i=1}^{n} \sigma_i \phi\left(f_j(x_i)\right) \right| \right]$$

$$\overset{\sum_{j=1}^{p_{L-1}} 1 = p_{L-1}}{=\!=\!=\!=\!=\!=} p_{L-1} R_n\left(\phi \circ \overline{\mathcal{F}}_A^{L-1}\right)$$

Eq. (26) holds due to the fact that most widely used activation functions $\phi$ (e.g., standard sigmoid and rectifier) have non-negative outputs.

Therefore, for both fully connected layers and convolutional layers, $R_n\left(\mathcal{F}_A^{L-1}\right) \leq p_{L-1} R_n\left(\phi \circ \overline{\mathcal{F}}_A^{L-1}\right)$ uniformly holds. Further considering the Lipschitz property of $\phi$, we have:

$$R_n\left(\mathcal{F}_A^{L-1}\right) \leq 2 p_{L-1} L_\phi R_n\left(\overline{\mathcal{F}}_A^{L-1}\right) \tag{27}$$

According to using maximization principle of inner product in Eq. (25), the property of RA in Eq. (26) and Lipschitz property in Eq. (27) iteratively, and considering $p_l \leq p$, we can obtain:

$$R_n\left(\mathcal{F}_A^L\right) \leq (2 p L_\phi A)^{L-1} R_n\left(\overline{\mathcal{F}}_A^1\right) \tag{28}$$

According to (Bartlett & Mendelson, 2002), $R_n\left(\overline{\mathcal{F}}_A^1\right)$ can be bounded by:

$$R_n\left(\overline{\mathcal{F}}_A^1\right) \leq b A M \sqrt{\frac{\ln(3HW)}{n}} \tag{29}$$

where $b$ is a constant. Ultimately, substitute Eq. (29) into Eq. (28), we can obtain:

$$R_n\left(\mathcal{F}_A^L\right) \leq b M A \sqrt{\frac{\ln(3HW)}{n}} (p L_\phi A)^{L-1} \tag{30}$$

Furthermore, substitute Eq. (30) into Eq. (6) in Lemma 2. Since $p L_\phi A > 1$, we have the generalization upper bound:

$$\mathrm{err}_P(f) \leq \mathrm{err}_S(f) + \mathcal{O}\left( L_\ell \frac{(p L_\phi A)^L}{n^{\frac{1}{2}}} \right) + c \sqrt{\frac{\log(1/\delta)}{2n}} \tag{31}$$

where $c$ is the upper bound of $L_1$ loss. Since $L_1$ loss is 1-Lipschitz, then $L_\ell = 1$ and we obtain the result of Theorem 1.

## C   THE PROOF OF LEMMA 3

We consider (scalar or vector-valued) standard BIQA neural networks, of the form:

$$\mathbf{x} \mapsto W_L \sigma_{L-1} \left( W_{L-1} \sigma_{L-2} \left( \ldots \sigma_1 \left( W_1 x \right) \right) \right) \tag{32}$$

And $N_{W_b^r}$ denote the function computed by the subnetwork composed of layers $b$ through $r$. Then, according to the Definition 3, the Rademacher complexity can be upper bounded as:

$$R_{n,\boldsymbol{\eta}} = \frac{1}{n} \mathbb{E}_{\boldsymbol{\epsilon}} \sup_{N_{W_1^{L-1}}, W_L} \sum_{i=1}^m \epsilon_i W_L \sigma_{L-1} \left( N_{W_1^{L-1}} \left( \eta_i x_i \right) \right)$$

$$\leq \frac{1}{n\lambda} \log \mathbb{E}_{\boldsymbol{\epsilon}} \sup \exp \left( \lambda \sum_{i=1}^m \epsilon_i W_L \sigma_{L-1} \left( N_{W_1^{L-1}} \left( \eta_i x_i \right) \right) \right) \tag{33}$$

$$\leq \frac{1}{n\lambda} \log \mathbb{E}_{\boldsymbol{\epsilon}} \sup \exp \left( M_F(L) \cdot \left\| \lambda \sum_{i=1}^m \epsilon_i \sigma_{L-1} \left( N_{W_1^{L-1}} \left( \eta_i x_i \right) \right) \right\| \right)$$

where each parameter matrix $W_j$ has Frobenius norm at most $M_F(j)$. We further simplify the expression based on the Lemma 1 in (Golowich et al., 2018) with $g(\alpha) = \exp \left( M_F(L) \cdot \lambda \alpha \right)$ as:

$$\frac{1}{\lambda} \log \mathbb{E}_{\boldsymbol{\epsilon}} \sup_{f, \|W_{d-1}\|_F \leq M_F(L-1)} \exp \left( M_F(L) \cdot \lambda \left\| \sum_{i=1}^m \epsilon_i \sigma_{L-1} \left( W_{L-1} f \left( \mathbf{x}_i \right) \right) \right\| \right)$$

$$\leq \frac{1}{\lambda} \log \left( 2 \cdot \mathbb{E}_{\boldsymbol{\epsilon}} \sup_f \exp \left( M_F(L) \cdot M_F(L-1) \cdot \lambda \left\| \sum_{i=1}^m \epsilon_i f \left( \mathbf{x}_i \right) \right\| \right) \right) \tag{34}$$

where $f \left( x_i \right) = \sigma_{L-2} \circ N_{W_1^{L-2}} (\eta_i x)$. Repeating the process, we arrive at

$$R_{n,\boldsymbol{\eta}} \left( \mathcal{F}_A^L \right) \leq \frac{1}{n\lambda} \log \left( 2^L \cdot \mathbb{E}_{\boldsymbol{\epsilon}} \exp \left( M\lambda \left\| \sum_{i=1}^n \epsilon_i \eta_i \mathbf{x}_i \right\| \right) \right)$$

$$= \frac{1}{n\lambda} \log \left( 2^L \cdot \mathbb{E}_{\boldsymbol{\epsilon}} \exp \left( M\lambda Q \right) \right). \tag{35}$$

## D   THE PROOF OF THEOREM 2

Following Lemma 3, we have

$$R_{n,\boldsymbol{\eta}} \left( \mathcal{F}_A^L \right) \leq \frac{1}{n\lambda} \log \left( 2^L \cdot \mathbb{E}_{\boldsymbol{\epsilon}} \exp \left( M\lambda \left\| \sum_{i=1}^n \epsilon_i \eta x_i \right\| \right) \right) \tag{36}$$

where $n$ denotes the number of training instances, $\mathbf{x}_i$ denotes the $i$-th instance, $\epsilon_i$ is a Rademacher variable, $\lambda$ is a random variable, and $M$ satisfies that:

$$M = \prod_{j=1}^L M_F(j) \tag{37}$$

Let $Z = M \left\| \sum_{i=1}^n \epsilon_i \eta x_i \right\|$, as a random function of the $n$ Rademacher variables. Then we have:

$$R_{n,\boldsymbol{\eta}} \left( \mathcal{F}_A^L \right) \leq \frac{1}{n} \frac{1}{\lambda} \log \left\{ 2^L \mathbb{E} \exp(\lambda Z) \right\} \tag{38}$$

Note that:

$$\frac{1}{\lambda} \log \left\{ 2^L \mathbb{E} \exp(\lambda Z) \right\} = \frac{L \log(2)}{\lambda} + \frac{1}{\lambda} \log \{ \mathbb{E} \exp \lambda (Z - \mathbb{E}[Z]) \} + \mathbb{E}[Z] \tag{39}$$

For the third term in the right part of Eq. (39), by Jensen's inequality, we have

$$\mathbb{E}[Z] \leq M \sqrt{\mathbb{E}_{\boldsymbol{\epsilon}} \left\| \sum_{i=1}^n \epsilon_i \eta_i x_i \right\|^2} = M \sqrt{\sum_{i=1}^n \eta_i^2 \|x_i\|^2} \tag{40}$$

According to the property of Rademacher variable, we have

$$Z\left(\epsilon_1,\ldots,\epsilon_i,\ldots,\epsilon_n\right) - Z\left(\epsilon_1,\ldots,-\epsilon_i,\ldots,\epsilon_n\right) \leq 2M\eta_i\left\|x_i\right\|. \tag{41}$$

By the bounded-difference condition (Boucheron et al.), $Z$ is a sub-Gaussian with variance factor:

$$\frac{1}{4}\sum_{i=1}^n\left(2M\left\|x_i\right\|\right)^2 = M^2\sum_{i=1}^n\eta_i^2\left\|x_i\right\|^2 \tag{42}$$

Hence, we have

$$\frac{1}{\lambda}\log\{\mathbb{E}\exp\lambda(Z-\mathbb{E}Z)\} \leq \frac{\lambda M^2\sum_{i=1}^n\eta_i^2\left\|x_i\right\|^2}{2} \tag{43}$$

Take the value of $\lambda$ as:

$$\lambda = \frac{\sqrt{2\log(2)L}}{M\sqrt{\sum_{i=1}^n\eta_i^2\left\|x_i\right\|^2}} \tag{44}$$

Then, for the second term in the right part of Eq. (39), we have:

$$\frac{1}{\lambda}\log\{\mathbb{E}\exp\lambda(Z-\mathbb{E}Z)\} \leq \frac{\lambda M^2\sum_{i=1}^n\eta_i^2\left\|x_i\right\|^2}{2} = \frac{M\sqrt{\log(2)L}\sqrt{\sum_{i=1}^n\eta_i^2\left\|x_i\right\|^2}}{\sqrt{2}} \tag{45}$$

For the first term in the right part of Eq. (39), we have:

$$\frac{L\log(2)}{\lambda} = \frac{M\sqrt{\log(2)L}\sqrt{\sum_{i=1}^n\eta_i^2\left\|x_i\right\|^2}}{\sqrt{2}} \tag{46}$$

Finally, for Eq. (39), we can obtain:

$$\frac{1}{\lambda}\log\left\{2^L\mathbb{E}\exp\lambda Z\right\} \leq M(\sqrt{2\log(2)L}+1)\sqrt{\sum_{i=1}^n\eta_i^2\left\|x_i\right\|^2} \tag{47}$$

According to (Bell et al., 1946), the chi-squre divergence between training and test distribution $D\left(P_{\text{test}}\|P_{\text{train}}\right)$ can be computed by $D\left(P_{\text{test}}\|P_{\text{train}}\right) = \int\frac{P_{\text{test}}^2(x_j)}{P_{\text{train}}(x_j)} - 1$. Note that $\eta_i = P_{\text{test}}(x_i)/P_{\text{train}}(x_i)$ defined in Lemma 3, we have:

$$\begin{aligned}
\frac{1}{n}\sum_{i=1}^n\eta_i^2\left\|x_i\right\|^2 &= \frac{1}{n}\sum_{i=1}^n\frac{P_{\text{test}}^2(x_i)}{P_{\text{train}}^2(x_i)}\left\|x_i\right\|^2 \\
&\approx \lim_{n\to+\infty}\frac{1}{n}\sum_{i=1}^n\frac{P_{\text{test}}^2(x_i)}{P_{\text{train}}^2(x_i)}\left\|x_i\right\|^2 = \mathbb{E}_{x_j\sim P_{\text{train}}}\left[\frac{P_{\text{test}}^2(x_j)}{P_{\text{train}}^2(x_j)}\left\|x_j\right\|^2\right].
\end{aligned} \tag{48}$$

Therefore, by the Law of Large Number (Durrett, 2019), we can obtain:

$$\frac{1}{n}\sum_{i=1}^n\eta_i^2\left\|x_i\right\|^2 \leq D\left(P_{\text{test}}\|P_{\text{train}}\right) + 1 + o\left(\frac{1}{\sqrt{n}}\right). \tag{49}$$

Then, through substituting Eq. (49) into Eq. (47) and Eq. (38), we can obtain $R_{n,\boldsymbol{\eta}}\left(\mathcal{F}_A^L\right)$ for BIQA model as follows:

$$\begin{aligned}
R_{n,\boldsymbol{\eta}}\left(\mathcal{F}_A^L\right) &\leq \frac{1}{n}\frac{1}{\lambda}\log\left\{2^L\mathbb{E}\exp(\lambda Z)\right\} \\
&\leq \frac{1}{\sqrt{n}}M(\sqrt{2\log(2)L}+1)\sqrt{\sum_{i=1}^n\eta_i^2\left\|x_i\right\|^2} \\
&= M(\sqrt{2\log(2)L}+1)\sqrt{\frac{1}{n}\sum_{i=1}^n\eta_i^2\left\|x_i\right\|^2} \\
&\leq M(\sqrt{2\log(2)L}+1)\sqrt{D\left(P_{\text{test}}\|P_{\text{train}}\right) + 1 + o\left(\frac{1}{\sqrt{n}}\right)} \\
&\leq \mathcal{O}\left(\sqrt{L}M\cdot\sqrt{D\left(P_{\text{test}}\|P_{\text{train}}\right)+1}\right)M(\sqrt{2\log(2)L}+1)
\end{aligned} \tag{50}$$

Finally, we can substitute $R_{n,\boldsymbol{\eta}}\left(\mathcal{F}_A^L\right)$ into Lemma 2 and obtain the result in Theorem 2:

$$\text{err}_P(f) \leq \text{err}_S(f) + \mathcal{O}\left(\frac{L_\ell \sqrt{L}M \cdot \sqrt{D\left(P_{\text{test}} \| P_{\text{train}}\right) + 1}}{\sqrt{n}}\right) + c\sqrt{\frac{\log(1/\delta)}{2n}} \qquad (51)$$

## E  THE PROOF OF THEOREM 3 AND ITS COROLLARIES

### E.1  BETTI NUMBER

Here We show the formal definition of Betti numbers as follows:

**Definition 4** ((Bianchini & Scarselli, 2014))**.** *For any subset $S \subset \mathbb{R}^d$, there exist $d$ Betti numbers, denoted as $b_j(S), 0 \leq d-1$. Therefore, the sum of Betti numbers is denoted as $B(S) = \sum_{j=0}^{d-1} b_j(S)$. Intuitively, the first Betti number $b_0(S)$ is the number of connected components of the set $S$, while the $j$-th Betti number $b_j(S)$ counts the number of $(j+1)$-dimension holes in $S$.*

According to Definition 4, we can reasonably generalize the definition of Betti numbers-based complexity into regression setting for BIQA models as Definition 4 in the main text.

### E.2  THE PROOF OF THEOREM 3

We first give the following Lemma, based on which we present the proof of Theorem 3.

**Lemma 4** ((Bianchini & Scarselli, 2014))**.** *Let $\sigma : \mathbb{R} \to \mathbb{R}$ be a function for which there exist a Pfaffian chain $c = (\sigma_1, \cdots, \sigma_\ell)$ and $\ell + 1$ polynomials, $Q$ and $P_i$, $1 \leq i \leq \ell$, of degree $\beta$ and $\alpha$, respectively, such that:*

$$\frac{d\sigma_i(a)}{da} = P_i\left(a, \sigma_1(a), \ldots, \sigma_i(a)\right), \quad 1 \leq i \leq \ell \qquad (52)$$

$$\sigma(a) = Q\left(\sigma_1(a), \ldots, \sigma_\ell(a)\right) \qquad (53)$$

*Let $f_{\mathcal{N}}(x)$ be the function implemented by a neural network with $n$ inputs, one output, $l \geq 1$ hidden layers, and $h$ hidden units with activation $\sigma$. Then, $f_{\mathcal{N}}(x)$ is Pfaffian with complexity bounded by $(\overline{\alpha}, \beta, h\ell)$, where $\overline{\alpha} = (\alpha + \beta - 1 + \alpha\beta)l + \alpha\beta$, in the general case, and $\overline{\alpha} = (\alpha + \beta - 1)l + \alpha$, if, $\forall i, P_i$ does not depend directly on $a$, i.e., $P_i = P_i(\sigma_1(a), ..., \sigma_i(a))$.*

The proof of Theorem 3 is shown as follows:

We first show that the functions $f(x) \in \mathcal{F}_A^L$ are the Pfaffian functions with the complexity $((\alpha + \beta - 1 + \alpha\beta)(L-1) + \alpha\beta, \beta, h\eta)$, where $\mathcal{F}_A^L$ can contain both fully-connected layers and convolutional layers. Assume the Pfaffian chain which defines activation function $\phi(t)$ is $(\phi_1(t), \phi_2(t), \cdots, \phi_\eta(t))$, and then $s^l$ is constructed by applying all $\phi_i, 1 \leq i \leq \eta$ on all the neurons up to layer $l-1$, i.e., $f^l \in \overline{\mathcal{F}}_A^1, l \in \{1, \cdots, L-1\}$. As the first step, we need to get the degree of $f^l$ in the chain $s^l$. Since $\phi$ is a Pfaffian function and:

$$f^l = \frac{1}{p_{l-1}} \sum_{k=1}^{m_{l-1}} w_k \left(\phi\left(f_{k,1}^{l-1}\right) + \cdots + \phi\left(f_{k,p_{l-1}}^{l-1}\right)\right) \qquad (54)$$

we can obtain that $f^l$ is polynomial of degree $\beta$ in the chain $s^l$. Then, it remains to show that the derivative of each function in $s^l$, i.e.,

$$\frac{\partial \phi_t\left(f^l\right)}{\partial x_{i,j,k}} = \frac{d\phi_t\left(f^l\right)}{df^l} \frac{\partial f^l}{\partial x_{i,j,k}} \qquad (55)$$

This can be defined as a polynomial in the functions of the chain and the input. For average pooling, by iteratively using chain rule, we can obtain that the highest degree terms of $\frac{\partial f^l}{\partial x_{i,j,k}}$ are in the form of $\prod_{i=1}^{l-1} \frac{d\phi\left(f^i\right)}{df^i}$. Following the Lemma 4, we obtain the complexity of $f(x) \in \mathcal{F}_A^L$ is $((\alpha + \beta - 1 + \alpha\beta)(L-1) + \alpha\beta, \beta, h\eta)$.

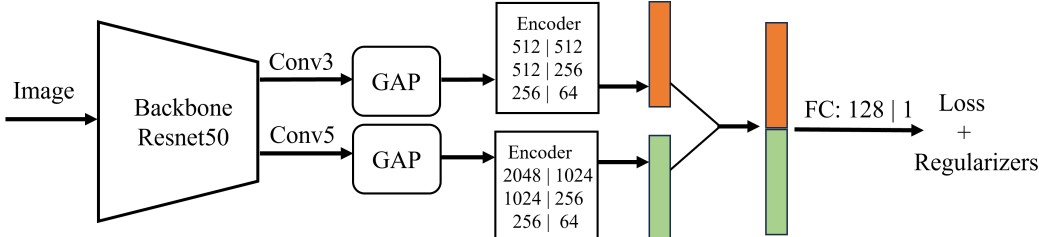

Figure 2: The design example that fuses multi-level features for BIQA model under suggestion (1). $C_1|C_2$ denotes a fully-connected (FC) layer mapping from dimension $C_1$ to $C_2$.

Table 6: Attributes of five typical IQA databases in experiments.

| Databases | Number | MOS Range | Distortion Type |
|---|---|---|---|
| TID2013 (Ponomarenko et al., 2015) | 3,000 | [0,9] | Synthetic |
| KADID-10k (Virtanen et al., 2014) | 10,125 | [1,5] | Synthetic |
| KonIQ-10k (Hosu et al., 2020) | 10,073 | [1,5] | Authentic |
| LIVE-C (Ghadiyaram & Bovik, 2015) | 1,162 | [0,100] | Authentic |
| CID2013 (Virtanen et al., 2014) | 480 | [0,100] | Authentic |
| SPAQ (Fang et al., 2020) | 11,125 | [0,100] | Authentic |

According to (Zell, 1999) and Definition 4, since $S_i$ is defined by 2 sign conditions (inequalities or equalities) on Pfaffian functions, and all the functions defining $S_i$ have complexity at most $((\alpha + \beta - 1 + \alpha\beta)(L-1) + \alpha\beta, \beta, h\eta)$, $B(S_i)$ can be upper bounded by $2^{d+h\eta(h\eta-1)/2} \times \mathcal{O}\left((d(\xi(L-1) + \beta(\alpha+1)))^{d+h\eta}\right)$, where $\xi = \alpha + \beta - 1 + \alpha\beta$ and $d = 3HW$.

Summing over all $i \in \{1, 2, \cdots, K\}$, we can get:

$$N\left(\mathcal{F}_A^L\right) \leq K2^{d+h\eta(h\eta-1)/2} \times \mathcal{O}\left((d(\xi(L-1) + \beta(\alpha+1)))^{d+h\eta}\right) \tag{56}$$

### E.3   THE PROOF OF COROLLARY 1

Since $\phi = \arctan(\cdot)$, the complexity of $\phi$ is $(3, 1, 2)$. According to the results in Theorem 3 and $3HW \leq 2h$, we have:

$$N\left(\mathcal{F}_A^L\right) \leq K2^{3HW+h(2h-1)} \times \mathcal{O}\left((3HW(L-1) + 3HW)^{3HW+2h}\right) \tag{57}$$

Since $\phi = \tanh(\cdot)$, the complexity of $\phi$ is $(2, 1, 1)$. According to the results in Theorem 3 and $3HW \leq h$, we have:

$$N\left(\mathcal{F}_A^L\right) \leq K2^{3HW+h(h-1)/2} \times \mathcal{O}\left((3HW(L-1) + 3HW)^{3HW+h}\right) \tag{58}$$

## F   THEORETICAL EXPLANATION FOR THREE SUGGESTIONS

For Suggestion (1), according to Theorems 1 and 2, as the level of image feature decreases, the generalization bound also decreases, and the generalization for quality perception becomes more excellent, which illustrates the significant role of low-level image features. According to Theorem 3, as the level of image feature increases, empirical error tends to decrease due to the greater representational power of BIQA networks, which illustrates the important role of high-level image features. Therefore, Suggestion (1) is theoretically valid.

For Suggestion (2), on the one hand, since the Lipschitz constant $L_\ell$ of the loss function is strongly related to the generalization bound, as shown in Eq. (31) in the proof of Theorem 1 and Eq. (11) in Theorem 2, an appropriate loss function can facilitate better generalization. Similar to the methods described in Category (2) in Section 6.3, Suggestion (2) promotes better generalization by improving the loss function with one regular term in Eq. (15). On the other hand, according to the analysis in

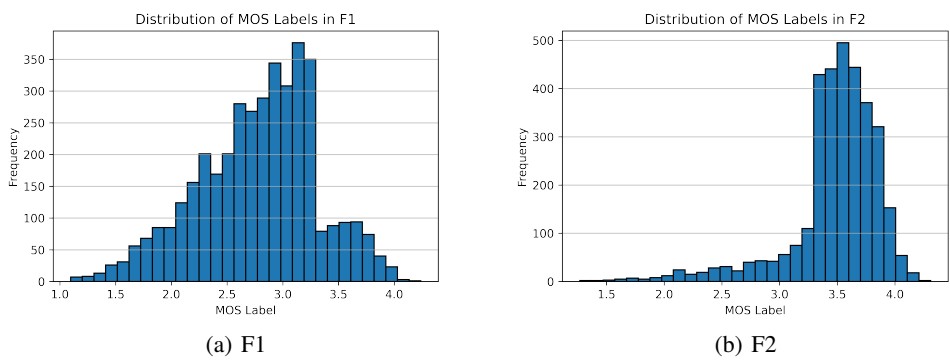

(a) F1       (b) F2

Figure 3: Distributions of MOS values in F1 (a) and F2 (b) on KonIQ-10k.

Section 4 and Section 5, we can conclude that there exists a conflict between good generalization and strong representation for BIQA networks with a restricted number of hidden units. In other words, when attempting to enhance the representation power of a BIQA network by increasing the learning level of quality perception features for MOS prediction, the model might have to incur the cost of weaker generalization capacity. This naturally leads to the following question: Can we reduce expected error and enhance the representation power of a BIQA network without increasing the complexity of learning quality perception features through an alternative approach? The answer is yes, and the regularization term in Suggestion **(2)** is a typical approach, which can enhance the representation power (i.e., reduce expected error) and keep good generalization simultaneously. Therefore, Suggestion **(2)** is theoretically valid.

For Suggestion **(3)**, according to the theoretical result in Eq. (8) Theorem 1, we can observe that a tighter weight parameter boundary $A$ may promote better generalization. This motivates us to apply a regularization penalty on the model parameters during the training process. Therefore, Suggestion **(3)** is theoretically valid.

Notably, the three suggestions are just examples to show that our proposed theoretical results and contributions can offer valuable insights for further exploration, which can be used as theoretical guidance or support for designs of the IQA models. Therefore, the theoretical results in the three proposed Theorems are the core contributions of this paper, rather than the three illustrative suggestions put forward.

## G EXPERIMENTAL DETAILS

### G.1 EXPERIMENTAL SETTINGS

**Implementation Details** Our experiments are conducted with the Pytorch library on two Intel Xeon E5-2609 v4 CPUs and four NVIDIA RTX 2080Ti GPUs. The batch size $B$ is set as 64. The training is conducted for just 100 epochs in total with SGD optimization. Meanwhile, we resize all the images into $256 \times 256$ and randomly center crop 10 sub-images to the size of $224 \times 224$. For the BIQA model with backbone of ResNet-18 in Table 1, Table 2 and Table 9, and BIQA model with backbone of ResNet-50 Table 3 and Table 4, we initialize the backbone by the pre-training weights obtained by classification task on ImageNet (Deng et al., 2009) before training. In the experiments of Table 3 and Table 4, we set $\mu = 10$ and $\nu = 0.01$. In the experiments of Table 9, we do not apply our three suggestions to the BIQA model with backbone ResNet-18 since this part of the experiment aims to study the impact of changes in the distribution from the training set to the test set on generalization performance. Since most experiments in this paper are cross-data experiments, we normalized the MOS labels of all datasets to [1,100] before training and testing. The intuitive construction process of different distributions of KonIQ-10k for the verification of Theorem 2 are shown in Figure 4

**Datasets** In this paper, we perform experiments on five representative authentically distorted image databases:

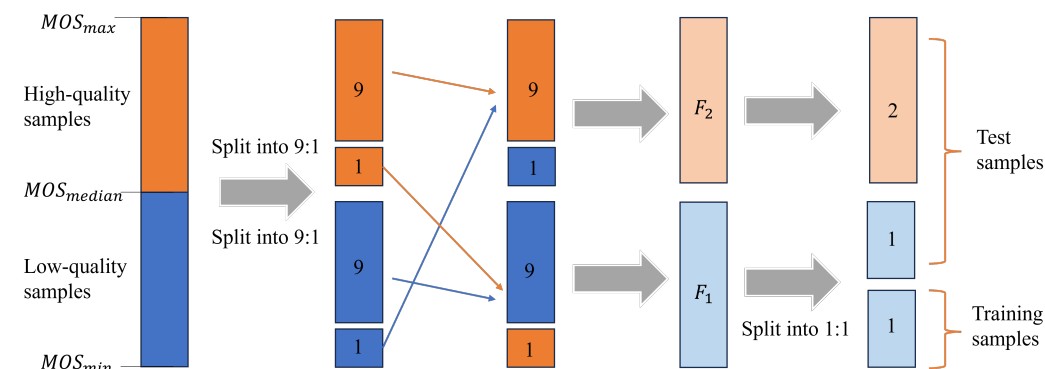

Figure 4: The intuitive construction process of different distributions of KonIQ-10k for the verification of Theorem 2 in experiments of Table 9 in the main text.

Table 7: Results on KonIQ-10k (Hosu et al., 2020) dataset.

| Models | PLCC | SRCC |
|---|---|---|
| DBCNN (Zhang et al., 2020) | 0.884 | 0.875 |
| MetaIQA (Zhu et al., 2020) | 0.887 | 0.850 |
| MUSIQ (Ke et al., 2021) (our run) | 0.917 | 0.904 |
| MUSIQ (Ke et al., 2021)+$S_2$+$S_3$ | **0.922** | **0.910** |

- KonIQ-10k (Hosu et al., 2020). It includes 10,073 images with authentic distortions chosen from YFCC100M (Thomee et al., 2016). Eight depth feature-based content or quality metrics are used in sampling process to ensure a wide and uniform distribution of image content and quality in terms of brightness, color, contrast and sharpness. And its quality is reported by MOS with the range of $[1, 5]$.

- LIVE-C (Ghadiyaram & Bovik, 2015). LIVE-C consists of 1,162 authentically distorted images captured from many diverse mobile devices. Each image was assessed on a continuous quality scale by an average of 175 unique subjects, and the MOS labels range in $[0, 100]$.

- TID2013 (Ponomarenko et al., 2015). This database contains 3,000 images, which are obtained from 25 reference images, 24 types of distortions for each reference image, and 5 levels for each type of distortion. The MOS labels range in $[0, 9]$

- SPAQ (Fang et al., 2020). SPAQ includes 11,125 images taken by 66 mobile phones, which contains a wide range of distortions during shooting, such as: sensor noise, blurring due to out-of-focus, motion blurring, over- or under-exposure, color shift, and contrast reduction. And the MOS labels range in $[0, 100]$.

- KADID-10k (Virtanen et al., 2014). It includes 81 pristine images, where each pristine image was degraded by 25 distortions in 5 levels. For each distorted image, 30 reliable degradation category ratings were obtained by crowdsourcing performed by 2,209 crowd workers. The MOS labels range in $[1, 5]$.

- CID2013 (Virtanen et al., 2014). CID2013 includes six image sets; on average, 30 subjects have evaluated 12–14 devices depicting eight different scenes for a total of 79 different cameras, 480 images, and 188 subjects. The MOS labels range in $[0, 100]$.

**Evaluation Metrics** We evaluate BIQA models by two typical metrics, including Pearson Linear Correlation Coefficient (PLCC) and Spearman Rank-order Correlation Coefficient (SRCC). In addition, the $L_1$ loss is also used as one metric to study the impact of the different learning level of quality perception features.

### G.2 DESIGN EXAMPLE AND OTHER RESULTS

**An Example of Our Suggestions for BIQA Model Design** Figure 2 is an example with Resnet-50 (He et al., 2016) as the backbone, which fuses 2 levels of features.

Table 8: Results on SPAQ (Fang et al., 2020) dataset.

| Models | PLCC | SRCC |
|---|---|---|
| FRIQUEE (Ghadiyaram & Bovik, 2017) | 0.830 | 0.819 |
| DBCNN (Zhang et al., 2020) | 0.915 | 0.911 |
| MUSIQ (Ke et al., 2021) (our run) | 0.912 | 0.909 |
| MUSIQ (Ke et al., 2021)+$S_2$+$S_3$ | **0.914** | **0.916** |

**Distribution of MOS values of F1 and F2 from KonIQ-10k**  In the Experimental Verification of Theorem 2, to study the impact of changes in the distribution from the training set to the test set on generalization performance, we have divided KonIQ-10k (Hosu et al., 2020) to two subsets with different distributions named F1 and F2. Figure 3 shows the distributions of MOS values in F1 and F2 on KonIQ-10k (Hosu et al., 2020).

**Combination of Advanced BIQA Network with Design Suggestions**  To further verify the rationality of our theoretical results and suggestions about the generalization ability of BIQA models, we train the advanced BIQA network MUSIQ (Ke et al., 2021) with Suggestion (**2**) ($\mu = 10$) and Suggestion (**3**)[1] ($\nu = 0.005$) on SPAQ and KonIQ-10k, the results of which are compared with that of the original MUSIQ (Ke et al., 2021). We record experimental results on KonIQ-10k and SPAQ on Table 7 and Table 8 respectively.

In addition, we compare our result with other baselines, including DBCNN (Zhang et al., 2020) and MetaIQA (Zhu et al., 2020) in the experiments on KonIQ-10k, and FRIQUEE (Ghadiyaram & Bovik, 2017) and DBCNN (Zhang et al., 2020) on SPAQ. The details of experimental settings are the same as described in MUSIQ (Ke et al., 2021). According to Table 7 and Table 8, the results of the combination of MUSIQ (Ke et al., 2021) and Suggestions (**2**) and (**3**) are better than that of original MUSIQ (Ke et al., 2021), which confirms the soundness of our theoretical conclusions and suggestions.

G.3  MORE DETAILED DESCRIPTION ABOUT EQ. (15)

In Eq. (15), $x_f$ denotes the extracted quality perception feature vector, and $R_B(x_f)$ represents the order (from smallest to largest) of the Euclidean distances between $x_f$ and the feature vectors of other image samples in the same batch $B$. $y$ denotes the ground truth MOS label for $x_f$, and $R_B(y)$ represents the order (from smallest to largest) of the absolute distances between $y$ and the MOS labels of other image samples in the same batch $B$. The absolute distances are computed by the absolute difference between two MOS scalars. cos refers to the cosine similarity. $Norm$ denotes the max-min normalization, for an original random variable $x$, that is:

$$Norm(x) = \frac{x - x_{\min}}{x_{\max} - x_{\min}} \tag{59}$$

where the maximum value of the cosine similarity in Eq. (15) is 1, and the minimum value f the cosine similarity in Eq. (15) is $\cos([1, 2, \ldots, B-1], [B-1, B-2, \ldots, 1])$.

**The second term in Eq. (15) means the consistency regularizer, corresponding to Suggestion (2)**, which directly minimizes empirical error by maintaining the consistency between the feature space and label space. The core idea stems from the conflict between strong representation and generalization in BIQA models revealed by Theorem 1 and Theorem 3. To avoid this conflict, instead of enhancing the model's representation power by increasing network depth, we focus on directly minimizing empirical error while maintaining good generalization. Therefore, the proposed consistency regularizer is an intuitive and effective choice. **The third term in Eq. (15) means the parameter regularizer, corresponding to Suggestion (3)**, where $\mathbf{W}_L$ denotes the weight parameter of the final prediction layer, and $|\cdot|_F$ is the Frobenius norm. $\mu$ and $\nu$ are the hyper-parameters.

According to Section 6.3, the proposed theorems can not only provide theoretical support for existing works, but also offer valuable insights for further exploration. Therefore, What we wish to emphasize is that: the proposed loss function in Eq. (15) just serves as the practical examples for the guidance

---

[1]Suggestion (**1**) is not considered here since the multi-scale features have already been incorporated in MUSIQ (Ke et al., 2021) during the training process.

of BIQA network design based on these theorems. In our future work, we will explore more comprehensive generalization theories for BIQA models and uncover practical values and insights to guide the design of future deep learning-based BIQA models.

## H Applicability of our Suggestions to Other Regression Tasks.

Although the network settings and loss functions may not be IQA-specific, the theoretical analysis and contributions in this paper have fully considered the task characteristics of the IQA domain. This is one of the core differences of this paper, distinguishing it from existing theoretical research, and it mainly includes the following two aspects.

- Focus on Regression Tasks in IQA: As stated in Section 2.2, most existing theoretical studies on deep neural networks focus on fully connected networks. Although some recent works have explored the generalization of CNNs, they are primarily applicable to classification tasks rather than regression tasks. However, the IQA task studied in this paper is a classic regression problem, making prior theoretical results for classification tasks unsuitable for the IQA tasks.

- Consideration of IQA-Specific Characteristics: As discussed in Section 6, this paper thoroughly considers the unique characteristics of IQA tasks: (1) quality perception information predominantly resides in low-level image features, and (2) effective representation learning of multi-level image features and distortion information is critical for the generalization of Blind IQA (BIQA) methods. In contrast, existing theoretical studies on general deep neural networks often overlook the role of low-level image features.

To illustrate this more intuitively, we conducted an experiment on another regression task, i.e. on the UTK-Face dataset (Zhang et al., 2017), where the task is to predict age from input facial images. Using ResNet-50 as the backbone, 80% of the data was used for training, and 20% for testing. The experimental setup followed Table 3, and the recorded MAE (Mean Absolute Error) results for $B$, $B + S_1$, $B + S_1 + S_2$, $B + S_1 + S_2 + S_3$ as follows:

Table 9: The MAE performances of our suggestions in this paper on the UTK-Face dataset (Zhang et al., 2017) in the age prediction task.

| Models | $B$ | $B + S_1$ | $B + S_1 + S_2$ | $B + S_1 + S_2 + S_3$ |
|--------|------|-----------|-----------------|------------------------|
| MAE | 4.96 | 5.03 | 4.88 | 4.91 |

From Table 9, we can observe that the three suggestions proposed in this paper perform poorly in the age prediction task. This is probably because the age prediction task primarily relies on high-level features about the age of the face in the input image. This indirectly confirms that the contributions of this paper are more relevant to IQA tasks, which differ significantly from other tasks.

