# OpenReview forum: "Understanding the Generalization of Blind Image Quality Assessment: A Theoretical Perspective on Multi-level Quality Features"
_ICLR.cc/2025/Conference — ICLR 2025 Conference Withdrawn Submission_

### Official Review · Reviewer_VqDJ · 2024-10-29

**Soundness:** 3
**Presentation:** 3
**Contribution:** 3
**Rating:** 6
**Confidence:** 2

**Summary:**

In this paper, the authors theoretically analyze the influence of multi-level image features on the generalization and quality perception abilities of CNN-based BIQA models. By establishing generalization bounds and employing Betti number-based analysis, they highlight the importance of low- and high-level features for model performance under distribution shifts. Their findings expose an inherent tension between generalization capacity and representational power, offering valuable insights for optimizing BIQA training.

**Strengths:**

1. This work provides the first theoretical establishment of a generalization boundary in the IQA domain, offering foundational guidance to improve BIQA model generalization.
2. Both key scenarios in IQA—distribution invariance and distribution shift—are thoroughly investigated.
3. Extensive experiments are conducted to empirically validate the theoretical findings.

**Weaknesses:**

1. Some symbols lack clear definitions in certain equations. For instance, $b$, $M$, $\delta$, and $L_{\phi}$ in Eqn. (7) and (8) are not well explained. Please explicitly define these symbols in the text preceding the equations to enhance clarity.
2. In Sec. 4.2, Theorem 1, the conclusion in (1) regarding the impact of increasing depth $L$ on the RA term's upper bound lacks rigor; increasing depth $L$ could also reduce empirical errors, potentially benefiting the generalization bound.  The authors are encouraged to clarify their conclusion under specific conditions. An additional discussion on the trade-off between the increase in the RA term and the potential decrease in empirical error would further enhance the analysis.
3. Theorem 2 suggests that the choice of loss function significantly affects the generalization capacity of BIQA models. It would be beneficial to analyze the advantages of loss functions like NIMA and Norm-in-Norm to further support Theorem 2. A brief comparative analysis of how different loss functions might influence the generalization bound derived in Theorem 2 is recommended.
4. While this paper focuses on CNN-based BIQA models, it would be useful to explore whether the conclusions apply to CNN-based general regression tasks, given that network settings or loss functions may not be IQA-specific. An additional discussion on the potential generalizability of the findings to other CNN-based regression tasks is recommended.

**Questions:**

Suggestions on the choice of loss function are advised, based on the insights from Theorem 2. It would be beneficial to test examples demonstrating how advanced loss functions might be more suitable for BIQA.

---

> ### Author Response · Authors · 2024-11-20
>
> Thank you for your affirmation and the valuable comments that help us improve our work. The following is a careful response and explanation about the weaknesses and questions. And we have submitted an updated PDF version of our improved paper.
> ### For Weakness 1
>
> Thank you for your thoughtful reminder. Although we have provided explanations for $b$, $M$, $\delta$, and $L_\phi$ in Lines 254, 248, 249, 231, 232, 259, respectively in the original pdf version, it may not be clear enough. We have made improvements in Lines 230-261 in the updated PDF version. To further enhance the clarity of our paper, we have explicitly defined these symbols in the text preceding the equations. Additionally, we have conducted a thorough review of the paper's notation to ensure that every symbol is clearly explained, thereby improving the overall readability and clarity of the manuscript.
>
> For the meaning of $b$, $M$, $\delta$, and $L \phi$, specifically:
> * $b$ is a constant independent of the other variables in Eq. (7), derived from the proof of Theorem 1.
> * $M$ represents the maximum value of the input image pixels.
> * $L_\phi$ is the Lipschitz constant of the activation function.
> * $\delta$ is a variable from Probability Theory that characterizes the degree of uniformity for the inequality in Eq. (8).
>
> ### For Weakness 2
> This is a misunderstanding. In fact, according to [1], the generalization bound is upper-bounded by the Rademacher complexities (RA term) and quantifies the difference between the expected loss and the empirical loss.
>
> In other words, the generalization bound refers to the gap between the test error and the training error, expressed as $err_P(f) - err_S(f)$, as shown below:
>
> $\operatorname{err}_P(f)- \operatorname{err}_S(f)\leq \mathcal{O} \left(\frac{\left(p L _\phi A\right)^{L}}{n^{\frac{1}{2}}} \right)+c \sqrt{\frac{\log (1 / \delta)}{2 n}}$
>
> Therefore, there is no issue of an indeterminate generalization bound caused by an increase in the RA term with larger L, while the $err_S(f)$ term decreases. Thus, Conclusion (1) in Theorem 1 holds true: With an increasing depth L, the upper bound of the RA term will increase, leading to a looser generalization bound. This indicates that relying solely on high-level quality perception features can negatively impact the generalization ability of BIQA models.
>
> In the updated PDF version, we have further emphasized this point in Lines 229 and 258 Section 4, clarifying that the generalization bound refers to $err_P(f) - err_S(f)$, not generalization bound refers to $err_P(f)-err_S(f)$, not $err_P(f)$. This reduces potential misunderstandings and further enhances the clarity of our paper.
>
> ### For Weakness 3 and Question 1
> Thank you for your valuable advice. In fact, as stated in Lines 318–321, based on the theoretical results and Conclusion (3) in Theorem 2, we have concluded that the generalization bound is linearly and positively correlated with the Lipschitz constant of the BIQA loss function. This suggests that the choice of loss function significantly impacts the generalization capability of the BIQA model, offering new insights into the success of existing methods like NIMA and Norm-in-Norm, as mentioned in Lines 409–411 in the original pdf version.
>
> However, calculating the Lipschitz constant for the loss functions used in NIMA and Norm-in-Norm is quite complex, which requires more time for in-depth exploration. This paper primarily focuses on theoretically exploring the impact of multi-level quality features on the generalization bound of BIQA models, while a comprehensive theoretical analysis of how different loss functions influence the generalization bound is another valuable topic for further investigation but is secondary to the primary scope of this work.
>
> We will continue to investigate this valuable issue raised by the reviewer regarding the theoretical impact of the loss functions used in NIMA and Norm-in-Norm, then we will explore this valuable topic about the theoretical influence of loss function on BIQA generalization more thoroughly in future researches and works.

---

> ### Author Response · Authors · 2024-11-20
>
> ### For Weakness 4
> Thank you for your valuable advice. Although the network settings and loss functions may not be IQA-specific, the theoretical analysis and contributions in this paper have fully considered the task characteristics of the IQA domain. This is one of the core differences of this paper, distinguishing it from existing theoretical research, and it mainly includes the following two aspects:
>
> * Focus on Regression Tasks in IQA: As stated in Lines 158–161 in the original pdf version, most existing theoretical studies on deep neural networks focus on fully connected networks. Although some recent works have explored the generalization of CNNs, they are primarily applicable to classification tasks rather than regression tasks. However, the IQA task studied in this paper is a classic regression problem, making prior theoretical results for classification tasks unsuitable for IQA.
>
> * Consideration of IQA-Specific Characteristics: As discussed in Section 6 and noted in Lines 64–67 and Lines 15–19 in the original pdf version, this paper thoroughly considers the unique characteristics of IQA tasks: (1) quality perception information predominantly resides in low-level image features, and (2) effective representation learning of multi-level image features and distortion information is critical for the generalization of Blind IQA (BIQA) methods. In contrast, existing theoretical studies on general deep neural networks often overlook the role of low-level image features.
>
> To illustrate this more intuitively, we conducted an experiment on another regression task, i.e. on the UTK-Face dataset [2], where the task is to predict age from input facial images. Using ResNet-50 as the backbone, 80% of the data was used for training, and 20% for testing. The experimental setup followed Table 3, and the recorded MAE （Mean Absolute Error） results for $B$, $B+S_1$, $B+S_1+S_2$,$B+S_1+S_2+S_3$ as follows:
> | Models  | $B$  |$B+S_1$ | $B+S_1+S_2$ |$B+S_1+S_2+S_3$ |
> |---------|---------|---------|---------|---------|
> | MAE | 4.96 | 5.03 | 4.88 | 4.91 |
>
> It can be observed that the three suggestions proposed in this paper perform poorly in the age prediction task, as this task primarily relies on high-level features. This indirectly confirms that the contributions of this paper are more relevant to IQA tasks, which differ significantly from other tasks. We have further emphasized this distinction in the discussion of Related Work in Section 2.2 (Generalization Bound in Deep Learning). We have also added the corresponding discussion and analysis in the Appendix H in Lines 1246-1280 in the updated PDF version.
>
> [1] Kakade S M, Sridharan K, Tewari A. On the complexity of linear prediction: Risk bounds, margin bounds, and regularization[J]. Advances in neural information processing systems, 2008, 21.
>
> [2] Zhang Z, Song Y, Qi H. Age progression/regression by conditional adversarial autoencoder[C]//Proceedings of the IEEE conference on computer vision and pattern recognition. 2017: 5810-5818.

---

### Official Review · Reviewer_M9u7 · 2024-11-02

**Soundness:** 2
**Presentation:** 2
**Contribution:** 2
**Rating:** 3
**Confidence:** 5

**Summary:**

This paper presents a theoretical analysis of multi-level image features in CNN-based BIQA models through three theorems. The findings reveal that low-level features enhance model generalization, high-level features boost representation power, and distribution differences between training and test sets affect generalization performance. Extensive experiments confirm these theoretical insights.

**Strengths:**

1. This paper investigates the role of multi-level image features in the generalization and quality perception ability of the CNN-based BIQA models from a theoretical perspective. Such theoretical research is rare in the field of IQA.
2. The theoretical findings are validated through both extensive experimental results and consistency with previous research observations in the field.

**Weaknesses:**

1. Limited contribution: The theoretical derivations in this paper primarily focus on analyzing existing consensus in the field using established theoretical frameworks, without presenting new insights, methods, or theoretical analysis approaches, thus lacking meaningful guidance for future work.
2. The paper's theoretical analysis focuses on model depth. Can the conclusions drawn from this effectively explain the role of multi-level feature fusion within the same network?
3. Theorem 1's significance is questionable as its core conclusions appear redundant with Theorem 2.
4. The derivation in Theorem 2 requires further explanation: 1) How is Equation 45 obtained? 2) The substitution of Equations 44 and 45 into Lemma 2 doesn't appear to yield Equation 46; please provide more detailed derivation steps. 3) There is an error in the definition of Z (missing ϵi).
5. Theorems 1 and 3 have different requirements for activation functions, potentially creating contradictions. For instance, ReLU is not a Pfaffian function and thus doesn't apply to Theorem 3, while tanh is not a non-negative function and doesn't apply to Theorem 1.
6. The meaning of Corollary 1 is unclear. It does not appear to have been explained in the paper.
7. The experimental section includes validation for Theorems 1 and 2, but lacks validation for Theorem 3.
8. In line 474, it is mentioned "According to the theoretical results of this work, we propose the Consistency Regularizer and Parameter Regularizer in the loss function Eq. (15)." However, both regularizers lack detailed explanations, and Eq. (15) is not thoroughly introduced.
9. There are some writing errors, such as in the y-axis of Fig. 1(b).

**Questions:**

N/A

---

> ### Author Response · Authors · 2024-11-20
>
> Thank you for the valuable comments that help us improve our work. The following is a careful response and explanation about the weaknesses. And we have submitted an updated PDF version of our improved paper.
> ### For Weakness 1
> Thank you for your valuable feedback. Although the theoretical analysis in this paper builds, to some extent, on previous theoretical research, we would like to clarify that our approach is distinct from existing theoretical frameworks, offering new insights and methods for IQA framework design.
>
> **On the one hand**, the core differences, adaptations, and contributions of this paper compared to existing theoretical research are as follows:
> * Focus on Regression Tasks: As stated in Lines 158-161 in the original pdf version, most existing theoretical studies on general deep neural networks concentrate on fully connected networks. While some works have explored the generalization of CNNs, these are primarily applicable to classification tasks, not regression tasks. However, IQA is a classic regression problem, and prior results on classification tasks are not directly applicable.
> * Consideration of IQA-Specific Characteristics: As discussed in Section 6 and mentioned in Lines 64-67 and Lines 15-19 in the original pdf version, our analysis explicitly considers the unique aspects of IQA tasks, namely: (1) Quality perception information predominantly resides in low-level image features. (2) Effective representation learning for multi-level image features and distortion information is critical to the generalization of Blind IQA (BIQA) methods.
> In contrast, existing theoretical research on deep neural networks often neglects the importance of low-level image features.
>
> **On the other hand**, our theoretical results present new insights and methods for IQA tasks. The contributions of this paper to practical BIQA model design and related research are comprehensively discussed in Lines 093-101 and Section 6 in the original pdf version. Key contributions include:
> * Revealing the Conflict Between Representation and Generalization in BIQA Models: We demonstrate that generalization ability tends to decrease while representation power increases as the level of image features rises. Consequently, as the level of quality perception features increases, the test error of BIQA networks may first decrease and then increase. This observation is validated in Figure 1 (c-d) in the Section 7 experiments.
> * Theoretical Explanation for Existing IQA Models: As outlined in Section 6.2, we provide theoretical explanations for the success of existing representative DL-based IQA models (e.g., CONTRIQUE, LIQA, MUSIQ, Hyper-IQA, Stair-IQA, NIMA, Norm-in-Norm). These insights are valuable for BIQA algorithm design. Notably, this is the first work to summarize these experiences from a theoretical perspective, laying a foundation for future BIQA algorithm development.
> * Providing Exemplary Suggestions Related to Proposed Theorems: As discussed in Section 6.3, our proposed theorems not only support existing works but also offer actionable insights for further exploration. The three proposed suggestions are typical examples. Extensive experiments validate the effectiveness and generality of these suggestions, demonstrating the correctness of the proposed theorems and their practical value for BIQA model design.
>
> To our best knowledge, this paper is the first theoretical analysis in the IQA field, and we believe it can mark a solid starting point from the theoretical perspective. In our future works, we will conduct deeper investigations into the generalization theory of more complex IQA models and provide even more valuable theoretical insights.
>
> ### For Weakness 2
> Yes, the conclusions drawn from the theoretical analysis in this paper effectively explain the role of multi-level feature fusion within a single network. In fact, as stated in Lines 176-177 in original PDF version, the BIQA networks examined in this paper consist of $L-1$ hidden layers for quality perception feature extraction and an output layer for MOS prediction. In other words, the depth of BIQA networks determines the level of image quality perception features. Therefore, our investigation and theoretical analysis for the role of multi-level image features are based on the depth of the quality feature extractor. This approach enhances the rigor of the theoretical derivation, as the depth of BIQA networks can be explicitly represented.

---

> > ### Author Response · Authors · 2024-11-20
> >
> > ### For Weakness 3
> > This is a misunderstanding. Theorem 1 and its core conclusions are not redundant with Theorem 2. This is primarily because the premises of Theorem 1 and Theorem 2 are fundamentally different. While Theorem 2 provides richer conclusions than Theorem 1, it also relies on more stringent and complex assumptions. Specifically:
> > * **Regarding the conclusions of the theorems**: Compared to Theorem 1, Theorem 2 focuses on the effect of distribution shifts, offering more comprehensive and detailed theoretical results.
> > * **Regarding the premises of the theorems**: Theorem 2 imposes stricter assumptions as prerequisites compared to Theorem 1. Specifically, **for Theorem 2**, the BIQA networks are modeled as a unified MLP, encompassing all sub-MLPs at different levels of quality perception representation. This unification is based on the observation that both convolutional and fully connected layers can be uniformly represented by Eq. (3) and Eq. (4), as explained in Lines 287–292 of the original PDF. In contrast, **for Theorem 1**, the theoretical results and derivations strictly adhere to the definition of CNN-based BIQA networks, as outlined in Eq. (2)–Eq. (5) in Section 3.
> >
> > In other words, Although Theorem 2 offers richer conclusions than Theorem 1, its assumptions are also more complex than Theorem 1. The advantage of Theorem 1 lies in providing more standard theoretical results and conclusions specifically tailored to CNN-based BIQA models. Therefore, Theorem 1 and its core conclusions are not redundant with Theorem 2.

---

> ### Author Response · Authors · 2024-11-20
>
> ### For Weakness 4
> Thank you for your reminder. We thoroughly reviewed the notation and proof process throughout our paper. In Lines 896-976 in the updated PDF version, we have corrected a few typographical errors and provided additional, more detailed steps in the proof of Theorem 2 to enhance the clarity and readability of the theoretical analysis.
>
> **For (1) and (2)**, we have provided more detailed derivations in the main text to help readers better understand how Equation (45) (in the original PDF version) is derived and how Equation (46) (in the original PDF version) is obtained by substituting Equations (44) and (45) (in the original PDF version) into Lemma 2.
>
> Specifically, based on the front part of the proof for Theorem 2, from Eq. (39), we can derive:
>
> $\frac{1}{\lambda}\log\\{2^L \mathbb{E} \exp \lambda Z\\} \leq M(\sqrt{2 \log (2) L}+1) \sqrt{\sum _{i=1}^{n} \eta _i^2 \left\|\|x _i\right\|\|^2}$
>
> According to [1], the chi-squre divergence between training and test distribution $D\left(P_{\text{test}} \|\| P_{\text{train}}\right)$ can be computed by $D\left(P_{\text{test}} \|\| P_{\text{train}}\right)=\int \frac{P_{\text{test}}^2\left(x_j\right)}{P_{\text{train}}\left(x_j\right)}-1$. Note that $\eta_i = P_{\text{test}}(x_i) / P_{\text{train}}(x_i)$ defined in Lemma 3, we have:
>
> $\frac{1}{n} \sum _{i=1}^{n} \eta _i^2 \left\|\|x _i\right\|\|^2 $
>
> $= \frac{1}{n} \sum _{i=1}^{n} \frac{P _{\text{test}}^2(x_i)}{P _{\text{train}}^2(x _i)} \left\|\|x _i\right\|\|^2 $
>
> $\approx  \lim _{n \rightarrow +\infty} \frac{1}{n} \sum _{i=1}^{n} \frac{P _{\text{test}}^2(x_i)}{P _{\text{train}}^2(x _i)} \left\|\|x _i\right\|\|^2 $
>
> $= \mathbf{E} _{x _j \sim P _{\text{train}}}\left[\frac{P _{\text{test}}^2\left(x _j\right)}{P _{\text{train}}^2\left(x _j\right)}\left\|\|x _j\right\|\|^2\right].$
>
> Therefore, by the Law of Large Number [2], we can obtain:
>
> $\frac{1}{n} \sum_{i=1}^{n} \eta_i^2 \left\|\|x_i\right\|\|^2 \leq  D\left(P_{\text {test}} \|\| P_{\text {train}}\right)+1+o\left(\frac{1}{\sqrt{n}}\right).$
>
> Then, through substituting Eq. (49) into Eq. (47) and Eq. (38), we can obtain $R_{n, \boldsymbol{\eta}}\left(\mathcal{F}_A^L\right)$ for BIQA model as follows:
>
> $R _{n, \boldsymbol{\eta}}\left(\mathcal{F} _A^L\right) \leq \frac{1}{n} \frac{1}{\lambda} \log \\{ 2^L \mathbb{E} \exp (\lambda Z) \\}$
>
> $\leq \frac{1}{\sqrt{n}} M(\sqrt{2 \log (2) L}+1) \sqrt{\sum _{i=1}^{n} \eta _i^2 \left\|\|x _i\right\|\|^2} $
>
> $=  M(\sqrt{2 \log (2) L}+1) \sqrt{\frac{1}{n}\sum _{i=1}^{n} \eta_i^2 \left\|\|x _i\right\|\|^2}$
>
> $\leq  M(\sqrt{2 \log (2) L}+1) \sqrt{D\left(P _{\text {test}} \|\|  P _{\text {train}}\right)+1+o\left(\frac{1}{\sqrt{n}}\right)} $
>
> $\leq   \mathcal{O}\left( \sqrt{L} M \cdot \sqrt{D\left(P _{\text{test}} \|\| P _{\text{train}}\right)+1}\right)M(\sqrt{2 \log (2) L}+1) $
>
> Finally, we can substitute $R_{n, \boldsymbol{\eta}}\left(\mathcal{F}_A^L\right)$ into Lemma 2 and obtain the result in Theorem 2 as follows:
>
> $\operatorname{err}_P(f) \leq \operatorname{err}_S(f) +\mathcal{O}\left(\frac{L _{\ell} \sqrt{L} M \cdot \sqrt{D\left(P _{\text{test}} \|\| P _{\text{train}}\right)+1}}{\sqrt{n}}\right) + c\sqrt{\frac{\log (1 / \delta)}{2n}}% \right)$
>
>
> **For (3)**, thank you for pointing this out. Yes, the definition of $Z$ was missing $\epsilon_i$ in the original PDF version. We have now corrected it to $ Z = M \left\|\|\sum_{i=1}^n \epsilon_i \eta x_i\right\|\| $
>
> [1] Bell R P, Everett D H, Longuet-Higgins H C. Kinetics of the base-catalysed bromination of diethyl malonate[J]. Proceedings of the Royal Society of London. Series A. Mathematical and Physical Sciences, 1946, 186(1007): 443-453.
>
> [2] Durrett R, Durrett R. Probability: theory and examples[M]. Cambridge university press, 2019.

---

> > ### Author Response · Authors · 2024-11-20
> >
> > ### For Weakness 5
> > In fact, although Theorems 1 and 3 impose different requirements on activation functions, they do not conflict with the conclusions of this paper; that is, there are no potential contradictions. The reasoning can be summarized as follows:
> >
> > * This paper mainly focuses on the theoretical exploration of the role of multi-level features in the generalization of BIQA models, while the choice of activation functions is not the primary focus of this study. In Theorems 1 and 3, the different activation functions are specified as conditions, aiming to mathematically express the nonlinear operations in CNN-based BIQA models, which helps derive more comprehensive theoretical results, including insights into how the concretized activation function impacts generalization or representation power. Therefore, these constraints on activation functions in Theorems 1 and 3 are merely to ensure rigorous theoretical proofs and do not alter the conclusions regarding the influence of multi-level quality features on the generalization of BIQA models.
> >
> > * In fact, ReLU can be seen as Pfaffian if the domain is broken into regions where $f(x)$ is either $0$ or $x$, with the transitions being well-defined. Therefore, there tend to be no contradictions between the conclusions of Theorems 1 and 3 for ReLU. In practice, there exist many activation functions that are both non-negative and can be considered Pfaffian under certain conditions, satisfying the assumptions of both Theorems 1 and 3 simultaneously. Examples include: **(1)** Sigmoid Function: The sigmoid function is smooth, bounded, positive, and satisfies the differential equation $f^{\prime}(x)=f(x)(1-f(x))$, making it a candidate for a Pfaffian function. **(2)** Softplus Function: The softplus function is smooth, non-negative, and satisfies $f^{\prime}(x)=\frac{e^x}{1+e^x}$. As it involves logarithmic and exponential terms, it can be represented as a Pfaffian function. In addition. **(3)** Exponential Linear Unit (ELU) is piecewise Pfaffian, since for $x>0$ and $x\leq 0$, it reduces to a linear function and involves $e^x$ respectively.
> >
> >
> >
> > * The conclusion derived from Theorem 1 can also be obtained under the more complex premise of Theorem 2. It is worth noting that Theorem 2 does not impose any restrictions on activation functions, which allows it to be compatible with the concretization conditions of the activation functions in Theorem 3. This demonstrates that the activation functions do not affect the core conclusions of this paper. Therefore, the concretization of different activation functions in Theorem 1 and Theorem 3 does not lead to potential contradictions.
> >
> >
> > The impact of different activation functions on the generalization ability of BIQA models is another important topic. In our future work, we will explore the theoretical effects of various activation functions on the generalization ability and representation power of deep learning-based BIQA models in greater depth.
> >
> > ### For Weakness 6
> > As stated in Lines 363-364 (in the original PDF version), we explained that Theorem 3 establishes an upper bound on the Betti numbers-based complexity for BIQA networks with general activation functions, and Corollary 1 provides a theoretical result derived from Theorem 3 for specific activation functions, such as arctan(x) and tanh(x). This corollary serves as an illustrative example of how Theorem 3 applies to particular activation functions, demonstrating its broader relevance and applicability.
> >
> > In Lines 362-372 in the updated PDF version, we have provided a more detailed explanation of the meaning and implications of Corollary 1 to enhance clarity and reader understanding.

---

> > > ### Author Response · Authors · 2024-11-20
> > >
> > > ### For Weakness 7
> > > This is a misunderstanding. We have verified **Theorem 3** in the experiment shown in Figure 1. In fact, **Figure 1 (a-d)** presents the experimental validation for both **Theorem 3** and **Theorem 1**.
> > >
> > > Specifically, **for the validation of Theorem 3**, as stated in Lines 373-377 in the original pdf version, we conclude that as the depth L increases, the Betti numbers-based complexity grows. This indicates that learning high-level quality perception features positively impacts the representation power of BIQA models, enabling them to better fit the training data and achieve smaller empirical errors. We validated this in **Figure 1 (a-b)** in the experiments section, where we observe that with an increase in L and the learning level of quality perception features, the empirical error of the BIQA model decreases. **For the validation of Theorem 1**, as stated in Lines 269-272 in the original pdf version, we conclude that with an increasing depth L, the upper bound of the RA term increases, leading to a looser generalization bound. This indicates that relying solely on high-level quality perception features can negatively impact the generalization ability of BIQA models, preventing them from achieving smaller test errors. We validated this in **Figure 1 (c-d)** in the experiments section, where we observe that as L and the learning level of quality perception features increase, the test error of the BIQA model decreases and then increases. **This also validates the conflict** between strong representation and generalization in BIQA models revealed in Section 6.1.
> > >
> > > In Lines 459-473 in the updated PDF version, **we have refined the description of this part in the experiments, emphasizing and highlighting the experimental validation for Theorems 1 and 3, as well as the revealed conflict**, to make the paper clearer and easier to understand, preventing any misunderstandings.
> > >
> > > ### For Weakness 8
> > > In Line 474 in the original pdf version, the mentioned **Consistency Regularizer** and **Parameter Regularizer** refer to the **second and third terms in Eq. (15)**, respectively. In Lines 445-452 in the updated PDF version, we have improved the description of this section by explicitly explaining the meanings of the Consistency Regularizer and Parameter Regularizer after Eq. (15).
> > >
> > > In the original PDF version, the explanations for Eq. (15) are provided in Lines 444-451. Specifically, $x_f$ denotes the extracted quality perception feature vector, and $R_B\left(x_f\right)$ represents the order (from smallest to largest) of the Euclidean distances between $x_f$ and the feature vectors of other image samples in the same batch $B$. $y$ denotes the ground truth MOS label for $x_f$, and $R_B\left(y\right)$ represents the order (from smallest to largest) of the absolute distances (computed as the absolute difference between two scalars) between $y$ and the MOS labels of other image samples in the same batch $B$. $\operatorname{cos}$ and $Norm$ refer to the cosine similarity and max-min normalization, where the maximum value is 1 and the minimum value is $\operatorname{cos}([1, 2, \dots, B-1], [B-1, B-2, \dots, 1])$. **The second term in Eq. (15) corresponds to Suggestion (2)**, which directly minimizes empirical error by maintaining the consistency between the feature space and label space. **The third term in Eq. (15) corresponds to Suggestion (3)**, where $\mathbf{W}_L$ denotes the weight parameter of the final prediction layer, and $| \cdot|_F$ is the Frobenius norm. $\mu$ and $\nu$ are the hyper-parameters.
> > >
> > > To make it easier for readers to understand Eq. (15), we have provided a more detailed explanation of Eq. (15) in the Appendix G.3 in Lines 1216-1245 of the updated PDF version. This enhances the readability and clarity of the paper, thereby improving the overall reading experience for the audience.
> > >
> > > ### For Weakness 9
> > > Thank you for your careful reminder. We have corrected the writing error in the y-axis of Fig. 1(b) (Test Loss → Empirical Loss). In addition, we have thoroughly reviewed and corrected any potential writing errors throughout the paper in the updated PDF version to ensure the accuracy of the paper's descriptions.

---

> ### Comment · Reviewer_M9u7 · 2024-11-25
>
> Based on the revised paper and the authors' response, I have decided to lower my rating for the following reasons:
>
> 1. **There are quite a few existing works (e.g. [1], [2])  studying generalization error bounds for regression tasks**, which contradicts the authors' claimed core contribution and their stated key differences from existing work. Furthermore, these relevant works have not been reviewed in the related work.
> 2. **The authors' contribution to the theoretical derivations is limited, and the reliability of Theorem 2 is questionable**:
> 	1. Eq. 7 in Theorem 1 of this paper originates from Theorem 2 in [3] (which is not properly cited). **Theorem 1 merely substitutes the upper bound of Rademacher complexity derived in [3] into Lemma 2** (the generalization error bound based on Rademacher complexity from [4]), with authors arbitrarily setting VC dimension d=3HW without further explanation.
> 	2. The proof of Theorem 2 raises reliability concerns. On page 18, lines 944-947 (and similarly in "Comment: For Weakness 4"), the authors state: "According to (Bell et al., 1946), the chi-square divergence between training and test distribution D(Ptest||Ptrain) can be computed by..." **Notably, [5] (Bell et al. 1946) is a chemistry-related paper, and it is implausible that it could be used to derive the chi-square divergence.**
> 	3. **Theorem 3 is almost identical to Theorem 3 in [3].** The only difference lies in the derivation process where the original uses "According to (Zell 1999), since Si is defined by K-1 sign conditions (inequalities or equalities) on Pfaffian functions," while this paper states "According to (Zell, 1999) and Definition 4, since Si is defined by 2 sign conditions (inequalities or equalities) on Pfaffian functions."
> 3. **The authors attempt to confuse the concepts of model depth (fundamentally a matter of model complexity) and hierarchical features to support their arguments.** These theorems essentially describe the bias-variance trade-off, which aligns with their observed phenomena: models with fewer layers have lower complexity and thus underfit the training data, while models with excessive layers lead to overfitting, performing well on training data but poorly on test data. However, this is clearly insufficient to directly justify that a large network using hierarchical features has characteristics of both large and small networks when training, as it fundamentally remains a large network. Therefore, the presented theory fails to provide a rigorous demonstration for the effectiveness of hierarchical features.
> 4. **This paper makes no methodological contributions, and its guidance merely reflects consensus in the field.**
>
> [1] Jakubovitz D, Giryes R, Rodrigues M R D. Generalization error in deep learning[C]//Compressed Sensing and Its Applications: Third International MATHEON Conference 2017. Springer International Publishing, 2019: 153-193.
>
> [2] Amjad J, Lyu Z, Rodrigues M R D. Regression with deep neural networks: generalization error guarantees, learning algorithms, and regularizers[C]//2021 29th European Signal Processing Conference (EUSIPCO). IEEE, 2021: 1481-1485.
>
> [3] Sun S, Chen W, Wang L, et al. On the depth of deep neural networks: A theoretical view[C]//Proceedings of the AAAI Conference on Artificial Intelligence. 2016, 30(1).
>
> [4] Bartlett P L, Mendelson S. Rademacher and Gaussian complexities: Risk bounds and structural results[J]. Journal of Machine Learning Research, 2002, 3(Nov): 463-482.
>
> [5] Bell R P, Everett D H, Longuet-Higgins H C. Kinetics of the base-catalysed bromination of diethyl malonate[J]. Proceedings of the Royal Society of London. Series A. Mathematical and Physical Sciences, 1946, 186(1007): 443-453.

---

> ### Author Response · Authors · 2024-11-28
>
> Thank you for your meticulous review and patient communication with us. Regarding the current concerns that still persist, we have provided an updated PDF version. So sorry,  due to time constraints, we can only upload **the new updated PDF version in the anonymous link：https://anonymous.4open.science/r/tmp-pdf-A6FE/2025_ICLR_IQA_update_2%20(1).pdf**, and we will make more detailed and comprehensive updates in the camera-ready version subsequently.
>
> ## For the new Concern 1
> This was an oversight on our part, and we have added the two articles on studying generalization error bounds for regression tasks to the related work section of our paper in the new updated PDF version in anonymous link **in Lines 157-161**. However, we would like to clarify that the core contribution of this paper and its key differences from existing work do not lie in the global generalization error bounds for regression tasks. As stated in Lines 159-161, we innovatively introduce a generalization bound specifically for regression networks in Image Quality Assessment (IQA) tasks. Our theoretical analysis fully takes into account the unique characteristics of IQA tasks, thereby effectively bridging the theoretical gap in the field of IQA. The unique characteristics of IQA tasks include: (1) the scale of training datasets is small due to high annotation costs; and (2) quality-aware features are closely related to low-level features, making IQA a low-level task. Therefore, the focus of this paper is on the role of high-level and low-level features in IQA networks, studying the generalization of IQA tasks, and providing a theoretical explanation for the design principles of existing IQA models.

---

> ### Author Response · Authors · 2024-11-28
>
> ## For the new Concern 2
> Thank you for your reminder. **We first provide an overall response, and then offer detailed replies to each specific point individually**.
>
> **For the overall response:** we acknowledge that our Theorems 1 and 3 are indeed inspired by existing research (as cited in our paper as Sun et al., 2016 in Lines 176 and 337). However, there are significant differences between our work and the issues and objectives explored in the cited papers. Sun et al. (2016) focused on classification tasks, while our work is dedicated to exploring the generalization of Image Quality Assessment (IQA) tasks and the theoretical interpretation of existing IQA model design principles. To achieve more precise citations, we have provided a more detailed citation of related works in the updated PDF version in anonymous link **in Lines 247 and 873**, especially in front of Theorem 1. Furthermore, Theorem 2 in our paper is entirely the result of our independent innovation, which is reasonable and rigorous. It not only reduces the exponential dependence of the generalization bound on $L$ in Theorem 1 to a linear dependence, but also effectively proposes a generalization bound related to the distribution difference between the training set and the test set. In our view, Theorem 2 makes an important contribution to the theoretical derivations and results.
>
> **For the point-by-point replies**, **for point (1)**, in our paper, prior to deriving the theoretical result Eq. 7 in Theorem 1, we first defined a CNN architecture tailored for the IQA task (as outlined in Eqs. 2-5). This differs from the architectures presented in the cited (Sun et al., 2016), resulting in differences in the theoretical conclusions regarding the Rademacher complexity compared to the cited work. Ultimately, based on the Rademacher complexity stated in Eq. 7 in our paper, we derived the generalization bound for the IQA task.
>
> Additionally, our proposed explanation for setting the VC dimension d=3HW in the generalization bound is provided in Lines 194-196 and Eq. 5, where $x$ represents the input image and 3×H×W denotes the dimensions of the input image.
>
> Moreover, we have given more meticulous consideration to the characteristics of regression networks in the field of Image Quality Assessment (IQA) by applying second-order pooling to quality feature extraction (as stated in the updated PDF version in anonymous link **in Lines 877-881**). This approach is recognized as an effective means of enhancing the performance of IQA models. Although we drew inspiration from the ideas presented in (Sun et al., 2016), the specifics of our proof differ from theirs.
>
> In addition, the derivation of the Theorem 2 in (Sun et al., 2016) only considers a part of the classification network. And if the entire classification network is considered, Eq. 13 in (Sun et al., 2016) does not hold. By contrast, there is no such limitation in Theorem 1 designed for BIQA regression model in our paper (as stated in the updated PDF version in anonymous link **in Lines 873-876**). Therefore, overall, although we are inspired by the proof framework from (Sun et al., 2016), our focus is on the IQA task. The investigated problems and the defined networks in our paper differ from those in (Sun et al., 2016), leading to distinct proof details. Furthermore, our proof is more rigorous, as it fully considers the structure of the regression network for IQA.
> Based on our theoretical results and analysis, we ultimately draw conclusions regarding the theoretical impact of low-level features on the performance of IQA models. To further highlight the differences from the proof conclusions presented in (Sun et al., 2016), we have refined the proof of Theorem 1 in the updated PDF version in anonymous link **in Lines 857-965**, emphasizing the distinctions from other works and providing more specific theoretical results and explanations.
>
> **For point (2)**, thank you for your reminder. We did not misquote, but it was indeed our oversight that led to a citation that is not precise enough. Specifically, the cited (Bell et al. 1946) are titled as "Kinetics of the base-catalysed bromination of diethyl malonate" is actually a journal book, and our intention was to cite the article at https://royalsocietypublishing.org/doi/epdf/10.1098/rspa.1946.0056, which is purely mathematical in nature and provides the definition of the chi-square divergence in Eq. 26. This paper is one part of "Kinetics of the base-catalysed bromination of diethyl malonate" and titled as "An invariant form for the prior probability inestimation problems". We have made the necessary correction in the updated PDF version in anonymous link **in Line 1049**. Therefore, the proof of our Theorem 2 is rigorous and reliable.

---

> > ### Author Response · Authors · 2024-11-28
> >
> > **For point (3)**, we acknowledge that our Theorem 3 was inspired by the derivation approach of Theorem 3 in (Sun et al., 2016). However, the problems we address are fundamentally different, and the resulting theoretical conclusions are also distinct. Specifically, Theorem 3 in (Sun et al., 2016) first introduces the definition of Betti number complexity for classification networks and then derives an upper bound on the representational power of such networks based on this definition. In our paper, we adopt this approach and, taking into account the characteristics of the IQA task, introduce a definition of Betti number complexity for IQA regression networks. Based on this definition, we derive an upper bound on the representational power of regression networks specifically tailored for the IQA task, which differs from the conclusions presented in (Sun et al., 2016). That is, Eq.12 in our work is different from Eq.18 in (Sun et al., 2016). Based on these theoretical results and analysis, we ultimately draw conclusions regarding the theoretical impact of high-level features on the performance of IQA models.
> >
> > In summary, the core objective of this paper is not to specifically study the generalization proof of neural networks, but rather to provide theoretical guarantees for the generalization of IQA tasks (taking into full consideration the characteristics of IQA tasks). We aim to explore the theoretical roles of low-level and high-level features in IQA models from a theoretical perspective and offer theoretical explanations and justifications for the success of existing IQA models. Although our Theorems 1 and 3 are inspired by proof approaches from prior work, we have thoroughly considered the task-specific characteristics of the IQA domain investigated in this paper. Furthermore, we have presented the theoretical impact of distribution differences between the training and test sets on the generalization bound. Our proof process is rigorous.

---

> > > ### Author Response · Authors · 2024-11-28
> > >
> > > ## For the new Concern 3
> > > We would like to clarify two aspects: (1) We are not focusing on large networks utilizing hierarchical features, nor have we provided rigorous theoretical proofs specifically for such large networks. Instead, we have theoretically investigated the respective roles of low-level and high-level features through a simple CNN-based IQA network. According to our theoretical results, low-level and high-level features complement each other in terms of model performance. Therefore, we suggest that IQA models should consider extracting multi-level features for training. (2) We have not confused the concepts of model depth and the level of features. As stated in Lines 176-177, in the defined CNN-based IQA network with depth $L$ in this paper, the quality perception feature extractor consists of $L−1$ hidden layers. This means that the depth $L$ of the defined CNN-based IQA network in this paper can indicate the level of the extracted features. Based on this, we can use model depth as a medium to theoretically study the roles of low-level and high-level features in model performance. Therefore, model depth and feature level are essentially two different concepts, but in our defined CNN-based IQA network, they are closely related. We have not confused these two concepts. In the updated PDF in anonymous link **in Lines 176-179**, we have further emphasized this point, stating that for the convenience of theoretical analysis, we use the depth of the quality perception feature extractor to reflect the level of features.
> > >
> > > ## For the new Concern 4
> > > We would like to clarify that this paper presents novel theoretical contributions and innovations, particularly in Theorem 2. Not only does it reduce the exponential dependence of the generalization bound on depth to a linear dependence in the context of IQA tasks, but it also explores the impact of distribution shifts between the training and test sets on the generalization bound. Furthermore, the core objective of this paper is not to solely investigate the generalization proofs of neural networks, but rather to provide theoretical guarantees for the generalization of IQA tasks, taking into full consideration the unique characteristics of IQA tasks. Although Theorems 1 and 3 are inspired by proof strategies from existing work, the problems we address are distinct. Our primary goal is not to focus exclusively on the generalization proofs of neural networks, but to offer theoretical assurances specifically for the generalization of IQA tasks. Our theoretical results not only elucidate the theoretical roles of low-level and high-level features in IQA models but also provide theoretical explanations and justifications for the success of existing IQA models.
> > >
> > > Therefore, we kindly request the reviewer to reconsider our score. We would be deeply grateful for your reconsideration.

---

### Official Review · Reviewer_qHP6 · 2024-11-03

**Soundness:** 3
**Presentation:** 3
**Contribution:** 3
**Rating:** 8
**Confidence:** 5

**Summary:**

This paper investigates the role of multi-level image features in the generalization and quality perception ability of the CNN-based BIQA models from a theoretical perspective. Specifically, three theorems are proposed, which can provide theoretical support for the enhanced generalization in existing BIQA methods.

**Strengths:**

This paper investigates the role of multi-level image features in the generalization and quality perception ability of BIQA models from a theoretical perspective.
Different generalization bounds for CNN-based BIQA networks are formulated.
The fundamental conflict between the generalization and representation abilities of CNN-based BIQA models is undercovered.

**Weaknesses:**

1.	This paper is pretty theoretical, which may give better understanding of the current BIQA models. However, the contributions of this paper to the practical BIQA model design and the relevant research should be highlighted.
2.	The theoretical derivations of the whole paper rely heavily on the existing theoretical research of general deep neural networks. The core differences/adaptions/contributions achieved in this paper (which are different from the existing research) should be well-explained.
3.	It seems that theoretical derivations only work for some specific architectures of BIQA models. The generalization to broader scope of BIQA networks should be discussed.
4.	Only limited BIQA models are selected to validate the theorems of the paper, and quantitative results in the paper are also limited.

**Questions:**

Following the above weaknesses, some questions are suggested to be answered.
1.	What are the contributions of this paper to the practical BIQA model design and the relevant research.
2.	What are the core differences/adaptions/contributions achieved in this paper, which are different from the existing research?
3.	What about other kinds of architectures beyond CNN, as more and more complicated model designs are involved. Nowadays, even LMMs are introduced for BIQA.
4.	More extensive experimental verifications are suggested to be given, for example, hand-crafted models like BMPRI, as well as DNN-based methods like RichIQA and StairIQA.

---

> ### Author Response · Authors · 2024-11-20
>
> Thank you for your affirmation and the valuable comments that help us improve our work. The following is a careful response and explanation about the weaknesses and questions. And we have submitted an updated PDF version of our improved paper.
>
> ### For Weakness 1 and Question 1
> The contributions of this paper to practical BIQA model design and related research are comprehensively discussed in Lines 093–101 and Section 6 in the original pdf version, with the main points summarized as follows:
> * Revealing the Conflict Between Strong Representation and Generalization in BIQA Models: Specifically, as the level of image features increases, generalization ability tends to decrease while representation power improves. Consequently, as the level of quality perception features rises, the test error of BIQA networks may first decrease and then increase. This conclusion is demonstrated in Figure 1(c–d) in Section 7 of the Experiments.
> * Providing a Theoretical Explanation for Existing IQA Models: As outlined in Section 6.2, the proposed theorems offer a theoretical explanation for the success of representative deep learning-based IQA models, such as CONTRIQUE, LIQA, MUSIQ, Hyper-IQA, Stair-IQA, and NIMA, among others. These insights provide valuable guidance for BIQA algorithm design. Notably, this paper is the first to summarize these findings from a theoretical perspective, laying a solid foundation for future BIQA algorithm development.
> * Providing Exemplary Suggestions Based on the Proposed Theorems: As stated in Section 6.3, the proposed theorems not only provide theoretical support for existing works but also offer meaningful insights for further exploration. The three suggestions proposed in this paper serve as typical examples. Through a series of experiments, we validated the effectiveness and generalizability of these suggestions, demonstrating the correctness of the theorems and their practical value in BIQA model design.
>
>
> In the updated PDF version, we further refined the explanation of the summarized contributions (listed in Lines 101–117), explicitly highlighting that our theoretical findings can assist BIQA model design from a feature-level perspective and explain the success of existing methods. This emphasizes the paper's contributions to practical BIQA model design and related research. Additionally, we improved the presentation of Section 6 to better underscore the practical value of the theoretical results for BIQA model development.
>
> ### For Weakness 2 and Question 2
>
> The core differences and and contributions of this paper, distinguishing it from existing research, lie in the following two aspects:
>
> * Focus on Regression Tasks in IQA: As stated in Lines 158–161 in the original pdf version, most existing theoretical studies on deep neural networks focus on BP  neural networks. Although some recent works have explored the generalization of CNNs, they are primarily applicable to classification tasks rather than regression tasks. However, the IQA task studied in this paper is a classic regression problem, making prior theoretical results for classification tasks unsuitable for IQA.
>
> * Consideration of IQA-Specific Characteristics: As discussed in Section 6 and noted in Lines 64–67 and Lines 15–19 in the original pdf version, this paper thoroughly considers the unique characteristics of IQA tasks: (1) quality perception information predominantly resides in low-level image features, and (2) effective representation learning of multi-level image features and distortion information is critical for the generalization of Blind IQA (BIQA) methods. In contrast, existing theoretical studies on general deep neural networks often overlook the role of low-level image features.
>
> To illustrate this more intuitively, we conducted an experiment on the UTK-Face dataset [1], where the task is to predict age from input facial images. Using ResNet-50 as the backbone, 80% of the data was used for training, and 20% for testing. The experimental setup followed Table 3, and the recorded MAE results for $B$, $B+S_1$, $B+S_1+S_2$,$B+S_1+S_2+S_3$ as follows:
> | Models  | $B$  |$B+S_1$ | $B+S_1+S_2$ |$B+S_1+S_2+S_3$ |
> |---------|---------|---------|---------|---------|
> | MAE | 4.96 | 5.03 | 4.88 | 4.91 |
>
> It can be observed that the three suggestions proposed in this paper perform poorly in the age prediction task, as this task primarily relies on high-level features. This indirectly confirms that the contributions of this paper are more relevant to IQA tasks, which differ significantly from other tasks. We have further emphasized this distinction in the discussion of Related Work in Section 2.2 (Generalization Bound in Deep Learning) in the updated PDF version.

---

> > ### Author Response · Authors · 2024-11-20
> >
> > ### For Weakness 3 and Question 3
> >
> > Since CNNs are classical neural networks for image processing, we chose CNN-based BIQA networks in this paper to rigorously derive and analyze the proposed theorems and theoretical results. In fact, the resulting theoretical conclusions and proposed suggestions are also applicable to other more complex network architectures, such as Vision Transformer-based BIQA models.
> >
> > To further elaborate on our viewpoint, we used ViT-base as the backbone and conducted cross-dataset experiments with the KADID-10k (train)/CLIVE (test) and KADID-10k (train)/CID2013 (test) database pairs. We recorded the results for $B$, $B+S_1$, $B+S_1+S_2$, and $B+S_1+S_2+S_3$ with the same experimental settings as Table 4.
> >
> > For the KADID-10k (train)/CLIVE (test) setup, the experimental results are as follows:
> > | Models  | PLCC | SRCC |
> > |---------|---------|---------|
> > | $B$ | 0.667 | 0.659 |
> > | $B+S_1$ | 0.681 | 0.675 |
> > | $B+S_1+S_2$ | 0.687 | 0.688 |
> > | $B+S_1+S_2+S_3$ | 0.692 | 0.686 |
> >
> > For the KADID-10k (train)/CID2013 (test) setup, the experimental results are as follows:
> > | Models  | PLCC | SRCC |
> > |---------|---------|---------|
> > | $B$ | 0.698 | 0.681 |
> > | $B+S_1$ | 0.725 | 0.720 |
> > | $B+S_1+S_2$ | 0.733 | 0.728 |
> > | $B+S_1+S_2+S_3$ | 0.736 | 0.731 |
> >
> > From the tables above, it can be observed that $B+S_1+S_2+S_3$ outperforms $B$, validating the effectiveness of the proposed suggestions and the correctness of the theoretical results presented in this paper.
> >
> > This work serves as our meaningful starting point for exploring the theory of generalization in IQA models. For more complex network architectures, including LMM-based BIQA models, we will conduct deeper explorations in future theoretical studies.
> >
> > ### For Weakness 4 and Question 4
> >
> > To validate the theorems of the paper with more BIQA models,we further conducted experiments on DBCNN [2] and Causal-IQA [3] models using KADID10k and CID2013 datasets, training on KADID10k and testing on CID2013. Similar to Tables 4 in the main text of our paper, we recorded the results for $B$, $B+S1$, $B+S_1+S_2$, $B+S_1+S_2+S_3$. The results show that our suggestions, based on our theoretical results, generally improve the generalization of BIQA models.
> >
> > The results of DBCNN are as follows:
> > | Models  | PLCC | SRCC |
> > |---------|---------|---------|
> > | $B$ | 0.677 | 0.654 |
> > | $B+S_1$ | 0.683 | 0.676 |
> > | $B+S_1+S_2$ | 0.705 | 0.711 |
> > | $B+S_1+S_2+S_3$ | 0.714 | 0.708 |
> >
> > The results of Causal-IQA are as follows:
> > | Models  | PLCC | SRCC |
> > |---------|---------|---------|
> > | $B$ | 0.734 | 0.715 |
> > | $B+S_1$ | 0.751 | 0.726 |
> > | $B+S_1+S_2$ | 0.763 | 0.744 |
> > | $B+S_1+S_2+S_3$ | 0.762 | 0.752 |
> >
> > We will continue to supplement the paper with additional experimental results. More quantitative results, including those for BMPRI, RichIQA, and StairIQA, will be presented in the camera-ready version.
> >
> > [1] Zhang Z, Song Y, Qi H. Age progression/regression by conditional adversarial autoencoder[C]//Proceedings of the IEEE conference on computer vision and pattern recognition. 2017: 5810-5818.
> >
> > [2] Zhang, W., Ma, K., Yan, J., Deng, D., Wang, Z. (2018). Blind Image Quality Assessment Using a Deep Bilinear Convolutional Neural Network. IEEE Transactions on Circuits and Systems for Video Technology.
> >
> > [3] Zhong, Y., Wu, X., Zhang, L., Yang, C., & Jiang, T. (2024). Causal-IQA: Towards the Generalization of Image Quality Assessment Based on Causal Inference. In Forty-first International Conference on Machine Learning.

---

> > > ### Comment · Reviewer_qHP6 · 2024-11-24
> > >
> > > Most of the previous concerns are addressed.
> > > If possible, some results with RichIQA and StairIQA will further strengthen my confidence, as these two are also typical methods which are not based on simple backbones. More complicated structures are involved in these networks.

---

> ### Author Response · Authors · 2024-11-28
>
> Thank you for acknowledging our work. Since StairIQA has publicly available code, we conducted experiments on StairIQA model using KADID10k, CLIVE and CID2013 datasets, training on  KADID10k and testing on CLIVE and CID2013 respectively. We recorded the results for $B$, $B+S_2$, and $B+S_2+S_3$ with the same experimental settings as Table 4.
>
> For the KADID-10k (train)/CLIVE (test) setup, the experimental results are as follows:
> | Models  | PLCC | SRCC |
> |---------|---------|---------|
> | $B$ | 0.651 | 0.662 |
> | $B+S_2$ | 0.655 | 0.671 |
> | $B+S_2+S_3$ | 0.660 | 0.667 |
>
> For the KADID-10k (train)/CID2013 (test) setup, the experimental results are as follows:
> | Models  | PLCC | SRCC |
> |---------|---------|---------|
> | $B$ | 0.705 | 0.712 |
> | $B+S_2$ | 0.713 | 0.724 |
> | $B+S_2+S_3$ | 0.728 | 0.721 |
>
> In our aforementioned experiments, the reason why we did not adopt the $B+S_1$ strategy is that StairIQA has already incorporated $S_1$, i.e., it has already implemented a multi-level feature fusion approach. As can be seen from the tables above, $B+S_2$ and $B+S_2+S_3$ perform better than $B$, which validates the effectiveness of the suggestions proposed in this paper and the correctness of our theoretical results.
>
> Furthermore, according to our research, the RichIQA work has not been published yet and there is no publicly available code, so it is unrealistic to reproduce it and conduct experiments in a short period of time. Therefore, we will try our best to include the reproduced RichIQA results in the camera-ready submission.
>
> Lastly, given that the reviewer believes that most of the concerns have been addressed, we kindly request the reviewer to consider further increasing the score if possible.

---

> > ### Comment · Reviewer_qHP6 · 2024-11-30
> >
> > The concerns are addressed.
> > I have raised my rating.

---

> > > ### Author Response · Authors · 2024-12-01
> > >
> > > We are deeply grateful for your review and sincerely appreciate your acknowledgment of our relentless efforts. Your feedback holds great significance for us.

---

### Official Review · Reviewer_zc7Q · 2024-11-04

**Soundness:** 2
**Presentation:** 2
**Contribution:** 2
**Rating:** 6
**Confidence:** 4

**Summary:**

The paper explores the generalization of CNN NR-IQA models and also investigates the roles of low- and high-level image features. It does establish theoretical generalization bounds, showing the effect of low-level  and high level features on generalization and representation capabilities.  The paper ends with 3 suggestions incorporated into a loss function that they claim to train CNN based IQA models better.

**Strengths:**

1. Theoretical analysis provides valuable insights into BIQA model performance in terms of generalizability, representation power, empirical results
2. This paper offers actionable insights drawn from theoretical results in the form of a new loss function to improve the training of CNN-based IQA models.
3. The results show that the method improves the training of BIQA methods. However, as highlighted in the weakness section, the experiment setup and results have some limitations.

**Weaknesses:**

1. Regarding cross-database experiment results in Table 1-4. In Tables 1-2, the results with $\mu, v =0$ should be provided. This will help gauge the usefulness of using the loss function proposed in Equation 15. In Table 4, it would be nice to have results in more than one cross dataset, or should authors justify using the KADID(train)/CID(test) database in the paper as to why the pair is a representative dataset?
2. Regarding results in Table 3, 7 the authors. The performance increase for the Resnet-50+ KonIQ scenario is much when incorporating the suggestions in the paper, while for MUSIQ+KonIQ, the performance boost is pretty low.  Do the authors have an intuition or explanation for this? Also, the results in Table 8 show negligible gains (probably statistically insignificant). Can this be interpreted as the method does not work well for transformer-based models?
3. Overall, the authors should also report RMSE in addition to SRCC and PLCC (as in many IQA/VQA papers). This should give an idea of improvement in predicted absolute values apart from the correlation.
4. Regarding the paper draft: Improving the paper's readability and organization of the text is needed.

**Questions:**

1. Please reply to Weaknesses #2
2. The paper ignores the use of the largest IQA (FLIVE, https://arxiv.org/pdf/1912.10088) database in the analysis and results. Could the authors explain the reasoning? The paper claims that generalization is limited by the small scale of the training IQA datasets, so leaving out the largest dataset needs some explanation.
3. How does the generalizability of the NR-IQA methods differ in two scenarios: real-life (UGC) distortions v/s synthetic distortions? Does training and testing in the same and different scenarios have any effect? Example: UGC(Train)- Synthetic(Test), UGC(Train)- UGC(Test) among others.

---

> ### Author Response · Authors · 2024-11-20
>
> Thank you for the valuable comments that help us improve our work. The following is a careful response and explanation about the weaknesses and questions. And we have submitted an updated PDF version of our improved paper.
> ## Responses to Weaknesses
> ### For Weakness 1
> Thank you for your valuable reminder.
>
> **Regarding the first point.** We have added experimental results for $\mu=0$ and $v=0$ to Tables 1 and 2, respectively, in the updated PDF version of the paper. The experimental settings remain consistent with those of Table 1 (with fixed $v=0.01$) and Table 2 (with fixed $\mu=10$).
>
> The current Table 1 has been updated to:
> |$\mu$  | 0 | 1 |5 | 10 | 15 |
> |---------|---------|---------|---------|---------|---------|
> | PLCC | 0.681 | 0.685 |0.702 | 0.713 | 0.708 |
> | SRCC | 0.685 | 0.679 | 0.692 | 0.697 | 0.684 |
> | RMSE | 13.37 | 14.16 | 13.05 | 12.21 | 12.74 |
>
> The current Table 2 has been updated to:
> |$v$  | 0 | 0.01 | 0.05 | 0.1 | 1 |
> |---------|---------|---------|---------|---------|---------|
> | PLCC | 0.709 | 0.713 |0.712 | 0.704 | 0.690 |
> | SRCC | 0.687 | 0.697 | 0.701 | 0.695 | 0.678 |
> | RMSE | 13.19 | 12.21 | 12.06 | 12.89 | 13.52 |
>
>
> From the updated Tables 1 and 2, it can be observed that setting $\mu=0$ or $v=0$ generally reduces the model's performance. This provides evidence, to some extent, of the effectiveness of the Consistency Regularizer and Parameter Regularizer in Eq. (15).
>
>
> In fact, $\mu=0$ and $v=0$ correspond to not adopting Suggestion (2) and Suggestion (3), respectively. The experimental effects of these suggestions have already been thoroughly discussed in Tables 3 and 4. Specifically:
> * When $\mu=0$, it implies not using the Consistency Regularizer (the second term in Eq. (15)) in the loss function, which corresponds to not adopting Suggestion (2) during training.
> * When $v=0$, it implies not using the Parameter Regularizer (the third term in Eq. (15)) in the loss function, which corresponds to not adopting Suggestion (3) during training.
>
> **Regarding the second point.** We use the KADID-10k (train)/CID2013 (test) database pair for cross-dataset experiments because CID2013 is one of the representative **authentically distorted** datasets, while KADID-10k is one of the representative **synthetically distorted** datasets. This setup effectively evaluates the generalization ability of BIQA methods.
>
> Additionally, to further validate the experiments comprehensively, we also use the KADID-10k (train)/CLIVE (test) and KADID-10k (train)/KonIQ-10k (test) database pairs for cross-dataset experiments, following the same experimental settings as in Table 4.
>
> For the KADID-10k (train)/CLIVE (test) setup, the experimental results are as follows:
> | Models  | PLCC | SRCC |
> |---------|---------|---------|
> | $B$ | 0.622 | 0.615 |
> | $B+S_1$ | 0.636 | 0.657 |
> | $B+S_1+S_2$ | 0.648 | 0.666 |
> | $B+S_1+S_2+S_3$ | 0.653 | 0.670 |
>
> For the KADID-10k (train)/KonIQ-10k (test) setup, the experimental results are as follows:
> | Models  | PLCC | SRCC |
> |---------|---------|---------|
> | $B$ | 0.641 | 0.664 |
> | $B+S_1$ | 0.663 | 0.681 |
> | $B+S_1+S_2$ | 0.688 | 0.697 |
> | $B+S_1+S_2+S_3$ | 0.702 | 0.685 |
>
> From the above results, it can be seen that, overall, the three exemplary suggestions proposed in this paper are effective. This is consistent with the conclusions drawn from Table 4, demonstrating the practical value of the theorems proposed in this paper.

---

> > ### Author Response · Authors · 2024-11-20
> >
> > ### For Weakness 2
> > In our opinion, this cannot be interpreted as the method being ineffective for transformer-based models. The main reason is as follows:
> >
> > As stated in Line 1187 in the original pdf version, in the experimental results for MUSIQ, Suggestion (1) is not considered in Tables 7 and 8 because multi-scale features are already incorporated into MUSIQ during the training process. Only Suggestions (2) and (3) are applied to MUSIQ in these tables, which is the primary reason for the relatively low performance boost of MUSIQ+$S_2+S_3$. Therefore, the results in Tables 7 and 8 do not fully reflect the overall effectiveness of the three suggestions. In fact, as stated in Lines 422–424 in the original pdf version, and supported by Conclusion (1) from Theorem 1 and Conclusion (1) from Theorem 3, the roles of low-level and high-level features are complementary. This provides theoretical evidence for the critical importance of Suggestion (1).
> >
> >
> > To further elaborate on our viewpoint, we used ViT-base as the backbone and conducted cross-dataset experiments with the KADID-10k (train)/CLIVE (test) and KADID-10k (train)/CID2013 (test) database pairs. We recorded the results for $B$, $B+S_1$, $B+S_1+S_2$, and $B+S_1+S_2+S_3$ with the same experimental settings as Table 4.
> >
> > For the KADID-10k (train)/CLIVE (test) setup, the experimental results are as follows:
> > | Models  | PLCC | SRCC |
> > |---------|---------|---------|
> > | $B$ | 0.667 | 0.659 |
> > | $B+S_1$ | 0.681 | 0.675 |
> > | $B+S_1+S_2$ | 0.687 | 0.688 |
> > | $B+S_1+S_2+S_3$ | 0.692 | 0.686 |
> >
> > For the KADID-10k (train)/CID2013 (test) setup, the experimental results are as follows:
> > | Models  | PLCC | SRCC |
> > |---------|---------|---------|
> > | $B$ | 0.698 | 0.681 |
> > | $B+S_1$ | 0.725 | 0.720 |
> > | $B+S_1+S_2$ | 0.733 | 0.728 |
> > | $B+S_1+S_2+S_3$ | 0.736 | 0.731 |
> >
> > From the Tables above, it can be seen that $B+S_1+S_2+S_3$ performs much better than $B$, which validates our opinion that the method can also work well for transformer-based models.
> >
> >
> > ### For Weakness 3
> > Thank you for your valuable advice! In addition to SRCC and PLCC, we have reported RMSE results in the main text's experimental section to provide a clearer understanding of the improvement in predicted absolute values.
> >
> > Specifically, Tables 1 and 2 have been updated to the versions included in our response to Weakness 1.
> >
> > The updated Table 3 is as follows:
> > | Models  | PLCC | SRCC | RMSE |
> > |---------|---------|---------|---------|
> > | $B$ | 0.857 | 0.865 | 7.005 |
> > | $B+S_1$ | 0.863 | 0.872 | 6.846 |
> > | $B+S_1+S_2$ | 0.872 | 0.866 | 6.703 |
> > | $B+S_1+S_2+S_3$ | 0.873 | 0.892 | 6.674 |
> >
> > The updated Table 4 is as follows:
> > | Models  | PLCC | SRCC | RMSE |
> > |---------|---------|---------|---------|
> > | $B$ | 0.771 | 0.689 | 12.91 |
> > | $B+S_1$ | 0.718 | 0.705 | 12.27 |
> > | $B+S_1+S_2$ | 0.725 | 0.726 | 11.68 |
> > | $B+S_1+S_2+S_3$ | 0.719 | 0.731 | 11.80 |
> >
> > The updated Table 5 is as follows:
> > | $F_1^{train}:(F_1^{test}:F_2^{test})$  | 1:(1:0) | 1:(1:0.5) | 1:(1:1) | 1:(1:2) | 1:(0:2) |
> > |---------|---------|---------|---------|---------|---------|
> > | PLCC | 0.818 | 0.775 | 0.746 | 0.723 | 0.695 |
> > | SRCC | 0.807 | 0.784 | 0.759 | 0.738 | 0.716 |
> > | RMSE | 8.308 | 8.621 | 9.248 | 9.874 | 10.122 |
> >
> >
> > From the above results, it can be observed that the trend in RMSE is essentially the opposite of that in PLCC and SRCC, which further confirms the correctness of the theoretical results presented in this paper.

---

> > > ### Author Response · Authors · 2024-11-20
> > >
> > > ### For Weakness 4
> > >
> > > Thank you for your valuable advice! We are continuously working to improve the readability and organization of the paper, including reducing the use of long and complex sentences, presenting the core contributions of the paper (i.e., the theoretical results of the three theorems) in a more accessible and straightforward manner, and adding more detailed discussions to clarify the relationships and potential value of the three theorems.
> > >
> > > ## Responses to Questions
> > > ### For Question 1
> > > Please reply to the response for Weaknesses 2
> > >
> > > ### For Question 2
> > >
> > > The main reasons we chose KADID-10k and KonIQ-10k for training rather than FLIVE in our experiments are as follows:
> > >
> > > * KADID-10k and KonIQ-10k are existing large-scale authentically distorted and synthetically distorted datasets, respectively.
> > > The samples from these two datasets each come from the same distribution, making KADID-10k and KonIQ-10k more representative for cross-dataset experiments to validate generalization. In contrast, FLIVE is a composite dataset created from multiple sources, including AVA, VOC, EMOTIC, and CERTH Blur, with a more complex and varied sample distribution, which complicates the validation of generalization in this paper.
> > > * The data scale of FLIVE (containing fewer than 40,000 samples) is close to that of KADID-10k and KonIQ-10k, each with over 10,000 samples, placing the data scales of all the three datasets within the same order of magnitudes. Although FLIVE is currently the largest IQA dataset, it is still relatively small in the field of computer vision. This limits the generalization of IQA models. True large-scale datasets should typically consist of millions of labeled samples to support deep learning training. For example, ImageNet contains 14 million labeled image samples.
> > >
> > > To conduct experiments with the FLIVE dataset, we used ResNet18 as the backbone and performed cross-database experiments based on the FLIVE (train)/CLIVE (test) database pair. The experimental settings were the same as those in Table 4. We recorded the results for $B$, $B+S_1$, $B+S_1+S_2$, and $B+S_1+S_2+S_3$ as follows:
> > > | Models  | PLCC | SRCC |
> > > |---------|---------|---------|
> > > | $B$ | 0.675 | 0.702 |
> > > | $B+S_1$ | 0.693 | 0.719 |
> > > | $B+S_1+S_2$ | 0.720 | 0.737 |
> > > | $B+S_1+S_2+S_3$ | 0.724 | 0.743 |
> > >
> > >
> > > From the above results, it can be seen that, overall, the three exemplary suggestions proposed in this paper are effective. This is consistent with the conclusions in Table 4, demonstrating the practical value of the proposed theorems. We will conduct more experiments based on FLIVE training in the camera-ready version to further validate the correctness of the theoretical results presented in this paper.
> > >
> > >
> > > ### For Question 3
> > > In our opinion, IQA models trained on authentically distorted images generally exhibit better generalization than those trained on synthetically distorted images. This is because the distortion types in authentically distorted images are much more complex than those in synthetically distorted images, a point that has been confirmed by many studies [1][2]. As a result, IQA models trained on synthetically distorted images tend to perform worse on authentically distorted images than on synthetically distorted images.
> > >
> > > Overall, when training in the same scenario, the performance of an IQA model is typically better than when training in different scenarios. However, this is not the main focus of our study. The proposed Theorem 2 in this paper focuses on the impact of sample distribution differences on the generalization of IQA models. We will further explore the theoretical implications of different scenarios on model generalization in future work.
> > >
> > > [1] Yang J, Lyu M, Qi Z, et al. Deep Learning Based Image Quality Assessment: A Survey[J]. Procedia Computer Science, 2023, 221: 1000-1005.
> > >
> > > [2] Yang P, Sturtz J, Qingge L. Progress in blind image quality assessment: a brief review[J]. Mathematics, 2023, 11(12): 2766.

---

> > > > ### Comment · Reviewer_zc7Q · 2024-11-23
> > > >
> > > > I thank the authors for providing extensive additional results. I have reviewed them and also looked at the reviewers' comments and responses.
> > > >
> > > > Overall, I would like to increase my score.

---

> > > > > ### Author Response · Authors · 2024-11-23
> > > > >
> > > > > Thank you so much for your review. We genuinely appreciate your acknowledgment and recognition of our tireless efforts. Your feedback means a great deal to us.

---

### Author Response · Authors · 2024-11-28

Dear ACs, SACs, PCs, and Reviewers,

After submitting our rebuttal, we have carefully prepared and now present to you the latest updated PDF version. However, due to time constraints, we have only uploaded the **newly updated PDF version to the anonymous link： https://anonymous.4open.science/r/tmp-pdf-A6FE/2025_ICLR_IQA_update_2%20(1).pdf**. We hereby specifically note that we kindly request ACs, PCs, and Reviewers to primarily refer to the new updated PDF version available at the anonymous link.

We sincerely apologize for any inconvenience this may have caused and deeply appreciate if you could take the time to review the new version. Your expert opinions are invaluable to us and will help us further refine our work.

Thank you once again for your hard work and valuable time. We look forward to your feedback.

Thank you very much!

---

### Public Comment · ~Alexander_Brown4 · 2024-12-02

All theory in this paper is assembled from theorems in multiple published studies. Theorem 1 is directly taken from Theorem 2 in [1], and Appendix B provides a proof that is identical to that in [1]. Similarly, Lemma 3 and Theorem 2 are sourced from Theorem 1 in [2], with Appendices C and D presenting proofs that are exactly the same as in [2]. Theorem 3 is directly taken from Theorem 3 in [1], and Appendix E.2 contains a proof that is also identical to the one in [1], including verbatim text.

It is concerning that above theory constitutes the main contribution of this paper, with the authors making virtually no theoretical innovations. They merely substituted symbols from the original texts, even retaining identical wording in the proofs. It is shocking that, despite other reviewers already pointing out these issues, this paper could still receive scores of 8 by **Reviewer qHP6**. This reflects a significant deterioration in both the ICLR's quality and the abilities of its reviewers.

[1] Sun S, Chen W, Wang L, et al. On the depth of deep neural networks: A theoretical view. Proceedings of the AAAI Conference on Artificial Intelligence. 2016; 30(1). https://arxiv.org/pdf/1506.05232

[2] Golowich, N., Rakhlin, A., & Shamir, O. (2018). Size-independent sample complexity of neural networks. https://arxiv.org/pdf/1712.06541

---

### Note · Authors · 2024-12-03

I have read and agree with the venue's withdrawal policy on behalf of myself and my co-authors.